# Bayesian calibration of firn densification models

Vincent Verjans[1], Amber A. Leeson[1], Christopher Nemeth[2], C. Max Stevens[3], Peter Kuipers Munneke[4], Brice Noël[4] and Jan Melchior van Wessem[4]

[1] Lancaster Environment Centre, Lancaster University, Lancaster, LA1 4YW, UK.
5    [2] Department of Mathematics and Statistics, Lancaster University, Lancaster LA1 4YF, UK
[3] Department of Earth and Space Sciences, University of Washington, Seattle, WA, USA
[4] Institute for Marine and Atmospheric research Utrecht, Utrecht University, Utrecht, the Netherlands

*Correspondence to*: Vincent Verjans (v.verjans@lancaster.ac.uk)

10   **Abstract.**

Firn densification modelling is key to understanding ice sheet mass balance, ice sheet surface elevation change, and the age difference between ice and the air in enclosed air bubbles. This has resulted in the development of many firn models, all relying to a certain degree on parameter calibration against observed data. We present a novel Bayesian calibration method for these parameters, and apply it to three existing firn models. Using an extensive dataset of firn cores from Greenland and Antarctica, 15   we reach optimal parameter estimates applicable to both ice sheets. We then use these to simulate firn density and evaluate against independent observations. Our simulations show a significant decrease (24 and 56%) in observation-model discrepancy for two models and a smaller increase (15%) for the third. As opposed to current methods, the Bayesian framework allows for robust uncertainty analysis related to parameter values. Based on our results, we review some inherent model assumptions and demonstrate how firn model choice and uncertainties in parameter values cause spread in key model outputs.

20   **1 Introduction**

On the Antarctic and Greenland ice sheets (AIS and GrIS), snow falling at the surface progressively compacts into ice, passing through an intermediary stage called firn. The process of firn densification depends on local conditions, primarily the temperature, the melt rate and the snow accumulation rate, and accurate modelling of densification is key to several applications in glaciology. Firstly, variability in firn densification affects altimetry measurements of ice sheet surface elevation 25   changes. Consequently, uncertainties in modelled densification rates have a direct impact on mass balance estimates, which rely on a correct conversion from measured volume changes to mass changes (Li and Zwally, 2011; McMillan et al., 2016; Shepherd et al., 2019). Errors in the firn related correction can lead to over- or underestimation of mass changes related to surface processes, and to misinterpreting elevation change signals as changes in mass balance and in ice flow dynamics. Secondly, firn models are used to estimate the partitioning of surface meltwater into runoff off the ice sheet, and refreezing 30   within the firn column, which strongly influences mass loss rates (van den Broeke et al., 2016). Model estimates of current and future surface mass balance of the AIS and GrIS are thus dependent on accurate models of firn evolution. And finally, the

densification rate determines the firn age at which air bubbles are trapped in the ice matrix. Knowing this age is crucial for precisely linking samples of past atmospheric composition, which are preserved in these bubbles, to paleo-temperature indicators, which come from the water isotopes in the ice (Buizert et al., 2014).

Firn densification has been the subject of numerous modelling studies over the last decades (e.g. Herron and Langway, 1980; Goujon et al., 2003; Helsen et al., 2008; Arthern et al., 2010; Ligtenberg et al., 2011; Simonsen et al., 2013; Morris and Wingham, 2014; Kuipers Munneke et al., 2015). However, there is no consensus on the precise formulation that such models should use. Most models adopt a two-stage densification process with the first stage characterising faster densification for firn with density less than a critical value, and then slower densification in the second stage. The firn-model intercomparison of

Lundin et al. (2017) demonstrated that, even for idealised simulations, inter-model disagreements are large in both stages. Firn compaction is driven by the pressure exerted by the overlying firn layers. Dry firn densification depends on numerous microphysical mechanisms acting at the scale of individual grains, such as grain-boundary sliding, vapour transport, dislocation creep and lattice diffusion (Maeno and Ebinuma, 1983; Alley, 1987; Wilkinson, 1988). Deriving formulations closely describing the densification of firn at the macroscale as a function of these mechanisms is challenging. Consequently, most

models rely on simplified governing formulations that are calibrated to agree with observations. The final model formulations have usually been tuned to data either from the AIS (Helsen et al., 2008; Arthern et al., 2010; Ligtenberg et al., 2011) or from the GrIS (Simonsen et al., 2013; Morris and Wingham, 2014; Kuipers Munneke et al., 2015), consisting of drilled firn cores from which depth-density profiles are measured. However, the calibration of firn densification rates to firn depth-density profiles requires the assumption of a firn layer in steady state. To overcome this limitation, some models have been calibrated

against other type of data such as strain rate measurements (Arthern et al., 2010; Morris and Wingham, 2014) or annual layering detected by radar reflection (Simonsen et al., 2013), but such measurements remain scarce and do not extend to firn at great depths below the surface. Ultimately, firn model calibration is an inverse problem that relies on using observational data to infer parameter values.

In this study, we adopt a Bayesian approach in order to address firn model calibration. This provides a rigorous mathematical framework for estimating distributions of the model parameters (Aster et al., 2005; Berliner et al., 2008). Bayesian inversion has been applied in several glaciological studies, and it has been demonstrated that this methodology improves our ability to constrain poorly known factors such as basal topography (Gudmundsson, 2006; Raymond and Gudmundsson, 2009; Brinkerhoff et al., 2016a), basal friction coefficients (Gudmundsson, 2006; Berliner et al., 2008; Raymond and Gudmundsson,

2009), ice viscosity (Berliner et al., 2008) and the role of the subglacial hydrology systems on ice dynamics (Brinkerhoff et al., 2016b). In the Bayesian framework, model parameters are considered as random variables for which we seek an *a posteriori* probability distribution that captures the probability density over the entire parameter space. This distribution allows not only to identify the most likely parameter combination, but also allows us to set confidence limits on the range of values in each parameter that is statistically reasonable. This enables us to quantify uncertainty in model results, to challenge the assumptions

inherent to the model itself and to assess correlation between different parameters. Calculations rely on Bayes' theorem (see Sect. 2.4 and Eq. (7)), but because of the high-dimensional parameter space and the non-linearity of firn models, solutions cannot be computed in closed form. As such, we apply rigorously designed Monte Carlo methods to approximate the target probability distributions efficiently. By exploiting the complementarity between the Bayesian framework and Monte Carlo techniques, we recalibrate three benchmark firn models and improve our understanding of their associated uncertainty.

## 2 Data and Methods

### 2.1 Firn densification data

In order to calibrate three firn densification models, we use observations of firn depth-density profiles from 91 firn cores (see Data Availability and Supplementary Information) located in different climatic conditions on both the GrIS (27 cores) and the AIS (64 cores) (Fig. 1). Using cores from both ice sheets is important since we seek parameter sets that are generally-applicable and not location-specific. We only consider dry densification since meltwater refreezing is poorly represented in firn models and wet-firn compaction is absent (Verjans et al., 2019). As such, we select cores from areas with low mean annual melt (<0.006 m w.e. yr$^{-1}$) but spanning a broad range of annual average temperatures (-55 to -20°C) and accumulation rates (0.02 to 1.06 m w.e. yr$^{-1}$). For each core, we use the depth-integrated porosity ($DIP$), also called firn air content. We calculate $DIP$ until 15 m depth ($DIP15$, Eq. (1)). For sufficiently deep measurements, we also calculate $DIPpc$, Eq. (2), taken below 15 m and until pore close-off depth ($z_{pc}$, where a density of 830 kg m$^{-3}$ is reached). These are the observed quantitative values used for the calibration:

$$DIP15 = \int_0^{15} \frac{\rho_i - \rho}{\rho_i} dz \tag{1}$$

$$DIPpc = \int_{15}^{z_{pc}} \frac{\rho_i - \rho}{\rho_i} dz \tag{2}$$

where $z$ (m) increases downwards, $\rho$ is the density of firn (kg m$^{-3}$) and $\rho_i$ is the density of ice (917 kg m$^{-3}$). In Eq. (2), we consider porosity only below 15 m to avoid dependency between $DIP15$ and $DIPpc$. We choose to use both $DIP15$ and $DIPpc$ in order to account for first- and second-stage densification. One of the cores has only a single density measurement above 15 m depth and thus its $DIP15$ value is discarded. We note that 48 cores are too shallow to reach $z_{pc}$ and so cores which do reach this depth provide a stronger constraint to the Bayesian inference method. This is sensible because these deep cores carry information about both stages of the densification process.

We use $DIP$ as the evaluation metric for the models because of the crucial role of this variable in both surface mass balance modelling and altimetry-based ice sheet mass balance assessments (Ligtenberg et al., 2014). We note that it is commonly used in firn model intercomparison exercises (Lundin et al., 2017; Stevens et al., 2020) and is a quantity of interest for field measurements (Vandecrux et al., 2018). Due to its formulation (Eq. (1) and (2)), $DIP$ represents the mean depth-density profile and thus is robust to the presence of individual errors and outliers in density measurements.

Observed firn density can be prone to measurement uncertainty, which previous studies point out is about 10%, though it is variable in depth and between measurement techniques employed (Hawley et al., 2008; Conger and McClung, 2009; Proksch et al., 2016). We outline our procedure to account for measurement uncertainty in Sect. 2.4.

We separate the dataset into calibration data (69 cores) and independent evaluation data (22 cores). The latter is selected semi-randomly; we ensure that it includes a representative ratio of GrIS-AIS cores and that it covers all climatic conditions, including an outlier of the dataset with high accumulation and temperature (see Supplementary Information). The resulting evaluation data has 8 GrIS and 14 AIS cores; 11 of the 22 cores extend to $z_{pc}$.

## 2.2 Climate model forcing

At the location of each core, we simulate firn densification under climatic forcing provided by the RACMO2.3p2 regional climate model (RACMO2 hereafter) at 5.5 km horizontal resolution for the GrIS (Noël et al., 2019) and 27 km for the AIS (van Wessem et al., 2018). Each firn model simulation consists of a spin-up by repeating a reference climate until reaching a firn column in equilibrium, which is followed by a transient period until the core-specific date of drilling. The reference climate is taken as the first 20-year period of RACMO2 forcing data (1960-1979 and 1979-1998 for the GrIS and AIS respectively). The number of iterations over the reference period depends on the site-specific accumulation rate and mass of the firn column (mass from surface down to $z_{pc}$). We ensure that the entire firn column is refreshed during the spin-up but fix the minimum and maximum number of iterations to 10 (200 years spin-up) and 50 (1000 years spin-up). We note that at 33 sites, the core was drilled before the last year of the reference climate and so the transient period is effectively a partial iteration of the spin-up period.

Results of the calibration would depend on the particular climate model used for forcing. We thus propagate uncertainty in modelled climatic conditions into our calibration of firn model parameters by perturbing the temperature and accumulation rates of RACMO2 with normally distributed random noise. Standard deviations of the random perturbations are based on reported errors of RACMO2 (Noël et al., 2019; van Wessem et al., 2018 – see more details in the Supplementary Information). By introducing these perturbations, uncertainty intervals on our parameter values encompass the range of values that would result from using other model-based or observational climatic input.

In addition to the climatic forcing, another surface boundary condition is the fresh snow density, $\rho_0$. At each site, the $\rho_0$ value is taken in agreement with the shallow densities measured in the corresponding core of the dataset. However, measurements of fresh snow density are highly variable (e.g. Fausto et al., 2018). We account for uncertainty in this parameter by adding normally distributed random noise with standard deviation 25 kg m$^{-3}$ to $\rho_0$ at every model time step (see Supplementary Information). We prefer this approach to the use of available parameterisations of $\rho_0$ (Helsen et al., 2008; Kuipers Munneke et al., 2015) to avoid any error in the fresh snow parameterisation to affect the calibration process.

## 2.3 Firn densification models

We use the Community Firn Model (Stevens et al., 2020) as the framework of our study because it incorporates the formulations of all three densification models investigated: HL (Herron and Langway, 1980), Ar (Arthern et al., 2010) and LZ (Li and Zwally, 2011). The Robin hypothesis (Robin, 1958) constitutes the fundamental assumption of HL, Ar and LZ. It states that any fractional decrease of the firn porosity, $\frac{\rho_i - \rho}{\rho_i}$, is proportional to an increment in overburden stress. This translates into densification rates depending on a rate coefficient $c$, assumed different for stage-1 and stage-2 densification:

$$\begin{cases} \frac{d\rho}{dt} = c_0 \, (\rho_i - \rho), & \rho \leq 550 \; kg \; m^{-3} \\ \frac{d\rho}{dt} = c_1 \, (\rho_i - \rho), & \rho > 550 \; kg \; m^{-3} \end{cases} \tag{3}$$

The formulations of the rate coefficients rely on calibration and thus differ between the three models investigated:

HL

$$\begin{cases} c_0 = \dot{b}^a k_0^* \exp\left(\frac{-E_0}{RT}\right) \\ c_1 = \dot{b}^b k_1^* \exp\left(\frac{-E_1}{RT}\right) \end{cases} \tag{4}$$

Ar

$$\begin{cases} c_0 = \rho_w \dot{b}^\alpha k_0^{Ar} g \exp\left(\frac{-E_c}{RT} + \frac{E_g}{RT_{av}}\right) \\ c_1 = \rho_w \dot{b}^\beta k_1^{Ar} g \exp\left(\frac{-E_c}{RT} + \frac{E_g}{RT_{av}}\right) \end{cases} \tag{5}$$

LZ

$$\begin{cases} c_0 = \beta_0 lz_a (273.15 - T)^{lz_b} \dot{b} \\ c_1 = \beta_1 lz_a (273.15 - T)^{lz_b} \dot{b} \end{cases} \tag{6}$$

$$with \begin{cases} \beta_0 = lz_{11} + lz_{12}\dot{b} + lz_{13}T_{av} \\ \beta_1 = \beta_0 \big(lz_{21} + lz_{22}\dot{b} + lz_{23}T_{av}\big)^{-1} \end{cases}$$

where $\dot{b}$ is the accumulation rate (m w.e. yr$^{-1}$), $T$ the temperature (K), $T_{av}$ the annual mean temperature, $R$ the gas constant, $g$ gravity and $\rho_w$ the water density (1000 kg m$^{-3}$). All remaining terms are model-specific tuning parameters. For $\dot{b}$, we use the mean accumulation rate over the lifetime of each specific firn layer because it better approximates the overburden stress than the annual mean (Li and Zwally, 2011). HL and Ar use Arrhenius relationships with activation energies ($E$ terms) capturing temperature sensitivity and exponents characterising the exponential proportionality of the rate coefficients to the accumulation rate. Originally, Herron and Langway (1980) inferred all values from calibration based on 17 firn cores, from which they inferred the values for the six free parameters (Table 1) of HL. In contrast, Arthern et al. (2010) fixed the accumulation exponents in advance ($\alpha = \beta = 1$) and took activation energies ($E_c, E_g$) from measurements of microscale mechanisms: Nabarro-Herring creep for $E_c$ and grain-growth for $E_g$. Still, they noted a mismatch with the activation energy fitting their data

best. The $k_0^{Ar}$ and $k_1^{Ar}$ parameters were tuned to three measured time series of strain rates collected in relatively warm and high accumulation locations of the AIS. Here, we consider all five $\alpha, \beta, k_0^{Ar}, k_1^{Ar}, E_g$ as free parameters (Table 1) but keep $E_c$ fixed because of its strong correlation with $E_g$; our use of monthly model time steps and depth-density profiles as calibration data is not suitable for differentiating effects of $\frac{E_g}{RT_{av}}$ and $\frac{E_c}{RT}$. Equation (6) shows that LZ has eight free parameters (Table 1),

all denoted by $lz$ in this paper. In contrast to our approach to Ar, we do not add additional accumulation rate exponents to $\dot{b}$ in Eq. (6) because the dependence of $c_0$ and $c_1$ on $\dot{b}$ also involves the coefficients $lz_{12}$ and $lz_{22}$ in the definition of $\beta_0$ and $\beta_1$. Li and Zwally (2011) performed their calibration of Eq. (6) against firn cores only from the GrIS. Later, Li and Zwally (2015) developed a densification model calibrated for Antarctic firn. The latter model uses the same governing equations as LZ for $c_0$ and $c_1$ but different formulations for $\beta_0$ and $\beta_1$ (Eq. (6)). Since one of the goals of this study is to find a densification

formulation applicable to firn in both the GrIS and AIS, we choose to apply our calibration method only to the formulations of $\beta_0$ and $\beta_1$ specified in Li and Zwally (2011) (Eq. (6)). However, in our results' analysis (Sect. 3), we also consider the performance of the Li and Zwally (2015) model on the AIS cores of our dataset.

## 2.4 Bayesian calibration

In our approach, the free parameters of the firn models are identified as the quantities of interest and we define this parameter set as $\theta$. Hereafter, 'original model values' refers to the values originally attributed by Herron and Langway (1980), Arthern et al. (2010) and Li and Zwally (2011) to their respective sets of free parameters $\theta$. The calibration process relies on Bayes' theorem (Eq. (7)) which allows to update a prior probability distribution $P(\theta)$ for $\theta$ based on observed data $Y$.

$$P(\theta|Y) = \frac{P(Y|\theta)P(\theta)}{P(Y)} \tag{7}$$

We use normal and weakly informative priors centred about the original model values so that the constraint of the prior on $P(\theta|Y)$ is minor (Table 1). As indicated by Morris and Wingham (2014), in HL and Ar, the values of the Arrhenius pre-exponential factors ($k_0^*, k_1^*, k_0^{Ar}$ and $k_1^{Ar}$) are correlated with their corresponding activation energies ($E_0, E_1$ and $E_g$). At a given temperature, a change of the value in the pre-exponential factor can be compensated by adjusting the activation energy to keep the densification rates constant. We express our *a priori* knowledge of these correlations in the prior distributions (see

Supplementary Information). No other pair of parameters in HL, Ar or LZ are clearly correlated *a priori*, but the calibration process captures *a posteriori* correlations by confronting the models with data. The data $Y$ consists of the observed $DIP15$ and $DIPpc$ values of the calibration data. The marginal likelihood, $P(Y)$, is a constant term independent of $\theta$ and does not influence the calibration. We use a normal likelihood function $P(Y|\theta)$, which quantifies the match of the modelled $DIP$ values with the observed:

$$P(Y|\theta) \propto \exp\left[\frac{-1}{2}(X_{15} - Y_{15})^T \Sigma_{15}^{-1}(X_{15} - Y_{15}) - \frac{1}{2}\left(X_{pc} - Y_{pc}\right)^T \Sigma_{pc}^{-1}\left(X_{pc} - Y_{pc}\right)\right] \tag{8}$$

where $X_{15}$ and $Y_{15}$ are vectors containing all modelled and observed values for the calibration data of $DIP15$ respectively, and similarly for $X_{pc}$ and $Y_{pc}$. We use diagonal covariance matrices $\Sigma_{15}$ and $\Sigma_{pc}$ with site-specific variances. The variances determine the spread allowed for the model outputs compared to the observed values and are calculated by taking 10% and

20% margins around $DIP15$ and $DIPpc$ measurements respectively. Allowing for such spread is necessary because multiple causes may lead to model-observation discrepancy such as firn model errors, measurement uncertainties and discrepancies induced by the random perturbations applied to RACMO2 forcing and to $\rho_0$. This particular form of the likelihood function assumes independence between model errors in $DIP15$ and in $DIPpc$, which is ensured by our calculation of $DIPpc$ only from 15 m depth to $z_{pc}$ (Eq. (2)). It also assumes normally distributed model errors with respect to the observed values. Both these

aspects were verified with preliminary assessments, along with our calculations for the covariance matrices $\Sigma_{15}$ and $\Sigma_{pc}$, as discussed in the Supplementary Information. The posterior distribution $P(\theta|Y)$ gives a probability distribution over the parameter space of a given model conditioned on the calibration data. In our case, with weakly informative priors (Table 1), the distribution $P(\theta|Y)$ is essentially governed by the likelihood function (Eq. (8)). We note here that extreme parameter combinations in the LZ model can lead to negative densification rates. In such cases, we set the modelled $DIP$ values to 0,

which leads to extremely low values for the likelihood and for the posterior probability of such parameter sets.

There is no analytical form of $P(\theta|Y)$ and we must investigate the parameter space to generate an ensemble of $\theta_i$ approximating $P(\theta|Y)$. Such an investigation is achieved efficiently using Markov Chain Monte Carlo methods. We apply the well-known Random Walk Metropolis (RWM) algorithm (Hastings, 1970) and summarize it in Fig. 2, on which we base the brief following

description. A given model (HL, Ar or LZ) starts with the original model parameter values and simulates firn profiles at all the calibration sites. Its $DIP15$ and $DIPpc$ results are compared with observations and the general performance of the model is quantified by the likelihood. From there and with the prior distributions assumed, the posterior probability is computed following Eq. (7). At this point, the RWM algorithm starts and the state of the chain, $\theta_i$ (Fig. 2a), is set to the original model values and its posterior probability is saved as $P(\theta_i|Y)$. It should be noted that the $i$ subscript designates the iteration number,

which is equal to 0 at this initial step. The RWM then proposes a new $\theta_i^*$ from a proposal distribution (Fig. 2b). For the latter, we use the symmetric multivariate normal (MVN) distribution which is centred about $\theta_i$. This implies that the random choice of $\theta_i^*$ depends only on the current state $\theta_i$ and on the proposal covariance in the MVN, $\Sigma_{prop}$, which is discussed below. Using the parameter combination $\theta_i^*$, the model simulates profiles at all calibration sites again (Fig. 2c) and $P(\theta_i^*|Y)$ is computed (Fig. 2d). From there, we either accept or reject the proposed $\theta_i^*$ in the ensemble approximating $P(\theta|Y)$. By using the

previously computed $P(\theta_i|Y)$, the probability of accepting $\theta_i^*$ depends on the ratio $P(\theta_i^*|Y)/P(\theta_i|Y)$ (Fig. 2e). The set saved in the ensemble (Fig. 2g) is $\theta_i^*$ if accepted or $\theta_i$ if $\theta_i^*$ was rejected. The saved set becomes the updated current status for the next iteration $\theta_{i+1}$ (Fig. 2a) with its associated posterior probability, $P(\theta_{i+1}|Y)$. The algorithm iterates this process and reaches

a final posterior distribution over $\theta$. The RWM has the property that the chain will ultimately converge to a stationary distribution that represents the posterior $P(\theta|Y)$. Thus, after a sufficiently high number of iterations of the algorithm, the ensemble of parameter sets is representative of $P(\theta|Y)$. We verify adequate convergence using a number of tests, which are shown in the Supplementary Information. The proposal covariance $\Sigma_{prop}$ must account for dependence between the different components of $\theta$, i.e. the value of one free parameter can influence the value of another free parameter for the model to reach a good match with the observed data. $\Sigma_{prop}$ can capture this dependence between parameters and, for optimality, it is updated every given number of iterations (100 in our study) using Eq. (9) (Rosenthal, 2010):

$$\Sigma_{prop} = \frac{2.38^2}{p} \Sigma_{cov} \qquad (9)$$

where $\Sigma_{cov}$ is the covariance matrix between the free parameters of the model at this stage of the iterative chain and $p$ is the number of free parameters.

From the posterior probability distributions, we can infer the Maximum a Posteriori (MAP) estimates of each model (MAP$_{HL}$, MAP$_{Ar}$, MAP$_{LZ}$). These are the modes of the multi-dimensional distributions over the space of free parameters and have been identified as the most likely sets by the RWM. The MAP estimates can be compared to the corresponding original model values of the parameters. The posterior distributions additionally incorporate the uncertainty in the parameter values. By performing posterior predictive simulations on the evaluation data, we can assess this remaining uncertainty (Gelman et al., 2013). More specifically, we can assume that a large (500) random sample of the ensemble of accepted $\theta$ is representative of the posterior distribution. As such, model results computed with all sets of this sample inform about model performance accounting for uncertainty. Intuitively, a large spread in results from 500 random samples would indicate a large range of possible sets for the free parameters and thus a high uncertainty in parameter values.

Since there is no analytical form of our posterior distributions, and to facilitate future firn model uncertainty assessments, we can approximate the posterior distributions with MVN distributions whose means and covariances are set to the posterior means and posterior covariance matrices of the calibration. This allows straightforward sampling of random parameter sets instead of relying on posterior samples of the MCMC. We provide information about the normal approximations and assess their validity in the Supplementary Information. Such normal approximations are asymptotically exact and are commonly applied to analytically intractable Bayesian posterior distributions (Gelman et al., 2013).

## 3 Results

We present the results of the calibration process after 15000 algorithm iterations and compare the MAP and original models' performances against the 22 evaluation cores. We also evaluate the uncertainty of the posterior distributions and compare performances between the different MAP models. All the evaluation simulations are performed without climatic and surface density noise in order to make the evaluation fully deterministic.

For HL and even more so for Ar, the posterior distributions for the parameters demonstrate some strong disagreements with the original values (Figs. 3a, 3b). The 95% credible intervals for each parameter (Table 1) incorporate 95% of the marginal probability density in the posterior. Two original parameter values of HL $(a, b)$ and three of Ar $(E_g, \alpha, \beta)$ lie in the tails of the

posterior distributions (Figs. 3a, 3b) and even outside these intervals in the case of $b, E_g, \alpha$ and $\beta$. This indicates that our analysis provides strong evidence against these original values. The strongest disagreements relate to the accumulation exponents of both models $(a, b, \alpha, \beta)$. In contrast, the original LZ values agree better with the posterior distribution and all lie within the 95% credible intervals (Table 1 and Fig. 3c). The posterior distributions show some strong correlation between certain pairs of parameters (Fig. 3). Notable examples are the pre-exponential factors and their corresponding activation energy

in HL and Ar, for which the posterior correlations are even stronger than in the prior distributions. The complete correlation matrices and a detailed analysis of all posterior correlation features are provided in the Supplementary Information.

We use the original models and the MAP estimates to simulate firn profiles at the evaluation sites and we compare $DIP$ results with the observed values. This is an effective way to assess possible improvements in parameter estimates reached through

our method since the evaluation sites were not used in the calibration process. The match between observations and the model is improved for MAP$_{HL}$ (Fig. 4a) and even more for MAP$_{Ar}$ (Fig. 4b), with the original Ar strongly underestimating $DIP$ values. These improvements translate into significantly reduced root mean squared errors (RMSE) in modelled values of both $DIP15$ (-24% for HL and -45% for Ar) and $DIPpc$ (-22% and -61%) (Table 2).

For LZ, the relative performance of the MAP$_{LZ}$ model for both $DIP15$ and $DIPpc$ is worse (+2% and +24% in RMSE) but

differences are of smaller magnitude (Table 2 and Fig. 4c). Parameter values of MAP$_{LZ}$ and the original LZ are closer, which explains more moderate differences in RMSE compared to HL and Ar. Comparing modelled and observed depth-density profiles of evaluation data illustrates the differences in performance visually (e.g. Fig. 5). Profiles of the original models of HL and Ar frequently lie outside the credible intervals of their respective MAP models. In contrast, profiles of MAP$_{LZ}$ and of the original LZ tend to be close together. At the climatic outlier of our evaluation data (DML in Fig. 5), improvements are

reached for the three MAP models (Figs. 5g, 5h, 5i). This demonstrates benefits of this method even at the limits of the calibration range. However, at a majority of the evaluation sites, the 95% credible intervals computed for the three models do not include the observed value (Fig. 4). This highlights that the governing equations of the models, which intend to capture densification physics, require improvement, and that parameter calibration in itself cannot overcome this shortcoming.

Compared to the original HL, MAP$_{HL}$ reaches improvements in $DIP15$ for 12 of the 22 evaluation cores and in $DIPpc$ for 5 of the 11 evaluation cores (Fig. 6a). Generally, MAP$_{HL}$ performs better at AIS sites and worse at GrIS sites. An analysis of the improvement of MAP$_{HL}$ as a function of climatic variables (Fig. 6a) shows that the original HL gives better results in a narrow range of $T_{av}$: from -30 to -25 °C. As such, the better performance at the GrIS evaluation sites of the original HL is likely due

to its parameterisation being better suited for the particular temperature range corresponding to the conditions of the latter sites. In contrast, $MAP_{HL}$ seems more appropriate for covering a wider range of climatic conditions. For Ar, the original model shows better performance than $MAP_{Ar}$ at few evaluation sites (6 for $DIP15$ and 2 for $DIPpc$) which are only in AIS and confined to low-accumulation conditions (Fig. 6b). This is counterintuitive given that Arthern et al. (2010) tuned the original Ar to measurements from high accumulation sites of the AIS. Finally, the original LZ performs better than $MAP_{LZ}$ at most GrIS sites (Fig. 6c), which is unsurprising given that its original calibration was GrIS-specific. Again, this seems related to the original LZ performing significantly better in the same narrow range of temperatures as for HL. In total, $MAP_{LZ}$ performs better for 10 of the 22 $DIP15$ and 4 of the 11 $DIPpc$ evaluation measurements.

As explained in Sect. 2.3, the original LZ model was developed for GrIS firn only (Li and Zwally, 2011) and later complemented by an AIS-specific model (Li and Zwally, 2015). We compute results at the AIS and GrIS evaluation sites using the Li and Zwally (2015) model for the AIS and the Li and Zwally (2011) model for the GrIS, so that both models are applied to the ice sheet for which they were originally developed. We call this pairing of models LZ dual and evaluate its general performance. The RMSE for $DIP15$ of LZ dual is slightly larger (+8 %) than that of $MAP_{LZ}$ and significantly larger (+38 %) for $DIPpc$ (Table 2). We note that the higher RMSE values of LZ dual are strongly affected by its densification scheme performing very poorly at the climatic outlier of the evaluation data, with conditions that are outside of the calibration range of Li and Zwally (2015).

We also compare MAP results with the IMAU firn densification model (IMAU-FDM), which has been used frequently in recent mass balance assessments from altimetry (Pritchard et al., 2012; Babonis et al., 2016; McMillan et al., 2016; Shepherd et al., 2019). IMAU-FDM was developed by adding two tuning parameters to both densification stages of Ar. All four extra-parameters are different for the AIS (Ligtenberg et al., 2011) and for the GrIS (Kuipers Munneke et al., 2015), thus also resulting in two separate models. On the evaluation data, its performance for $DIP15$ is slightly better than $MAP_{Ar}$ and $MAP_{LZ}$ but worse than $MAP_{HL}$, and its performance for $DIPpc$ is significantly worse than all three MAP models (Table 2).

To assess the uncertainty captured by the Bayesian posterior distributions, we compute results on the evaluation data with the 500 parameter sets randomly selected from each of the three posterior ensembles. For all three models, the average performance of their random sample is similar to the corresponding MAP performance, with a maximum RMSE change of 6% (Table 2). This demonstrates a low uncertainty in the optimal parameter combinations identified by calibration. Furthermore, the best performing 95[th] percentile of the random selection allows the construction of the uncertainty intervals shown in Figs 4, 5. Of the original models, LZ reaches the lowest RMSE values. Of all models, $MAP_{HL}$ performs best in $DIP15$ and $MAP_{Ar}$ in $DIPpc$ (Table 2). $MAP_{LZ}$ performs worse than the other MAP models even when accounting for uncertainty by using the 500-samples random selections (Table 2).

## 4 Discussion

This calibration method is potentially applicable to models of similar complexity in a broad range of research fields. We exploit it here to investigate the parameter space of HL, Ar and LZ, and to re-estimate optimal parameter values conditioned on observed calibration data; no further complexity is introduced since the number of empirical parameters remains the same. We
treat the accumulation exponents of Ar $(\alpha, \beta)$ as free parameters whereas Arthern et al. (2010) decided to fix their values to 1. Analogous to $a$ and $b$ in HL, these exponents capture the mathematical relationship between densification rates and the accumulation rate, used as a proxy for load increase on any specific firn layer. No physical argument favours a linear proportionality between densification and load increase and any prescribed value for these exponents is a choice of the model designer. Unlike Arthern et al. (2010), Herron and Langway (1980) previously inferred $a = 1$ and $b = 0.5$. Our calibration
data shows strong evidence against both these pairs of values; all four are in the extreme tails of the posterior distributions (Fig. 3a, 3b). Our results of stage-1 exponents $(a, \alpha)$ smaller than 1 indicate a weaker increase in densification rates with pressure than assumed in the original versions of Ar and HL. In firn, the load is supported at the contact area between the grains, which increases on average due to grain rearrangement (in stage-1) and grain growth. As such, firn strengthens in time and the actual stress on ice grains increases slower than the total load (Anderson and Benson, 1963). Morris and Wingham
(2014) incorporated this by including a temperature-history function, causing slower densification of firn previously exposed to higher temperatures. This is consistent with both grain rearrangement and grain growth because these processes are enhanced at higher temperatures (Alley, 1987; Gow et al., 2004). Lower values of the stage-2 exponents $(b, \beta)$ illustrate the larger strength of high-density firn with larger contact areas between grains. The difference in sensitivities of stage-1 and stage-2 densification to accumulation also holds in the LZ model, as illustrated by the posterior correlation between its free parameters.
The correlation coefficient between the accumulation-related parameters of both stages, $lz_{12}$ and $lz_{22}$, is significantly positive (0.74, Fig. S5). High values of $lz_{12}$ make $\beta_0$ more sensitive to $\dot{b}$ (Eq. (6)). However, $\beta_0$ appears in the numerator of the $\beta_1$ calculation (Eq. (6)) and higher values of $lz_{22}$ thus moderate the sensitivity of stage-2 densification to $\dot{b}$. As such, positively correlated $lz_{12}$ and $lz_{22}$ provide further evidence that stage-1 densification rates are more sensitive to accumulation rates. This example demonstrates how posterior correlations provide insights into model behaviour. The posterior correlations of all three
models are further discussed in the Supplementary Information.

In the IMAU model introduced in Sect. 3, tuning parameters have been added to Ar in order to reduce its sensitivity to accumulation rates (Ligtenberg et al., 2011; Kuipers Munneke et al., 2015). The calibration method presented in this study detects and adjusts for this over-sensitivity in Ar without the need for more tuning parameters in the governing densification
equations. The sensitivity of stage-1 densification to $\dot{b}$ can be computed from the derivative of the rate coefficient:

$$\frac{\partial c_0}{\partial \dot{b}} = \rho_w \, k_0^{Ar} \, g \, \exp\left(\frac{-E_c}{RT} + \frac{E_g}{RT_{av}}\right) \alpha \, \dot{b}^{\alpha-1} \tag{10}$$

Similarly, the derivative $\frac{\partial c_1}{\partial b}$ is obtained by replacing $k_0^{Ar}$ and $\alpha$ with $k_1^{Ar}$ and $\beta$. Our calibration process strongly favours smaller values of $\alpha$, $\beta$ and $E_g$ with respect to the original values (Fig. 3b). We can compare the magnitudes of the derivatives under the original Ar parameterisation and under the MAP parameterisation. The magnitudes vary for particular combinations of $T_{av}$ and $\dot{b}$. Under all the annual mean climatic regimes of our dataset, the MAP parameters result in a decreased sensitivity

of both stage-1 and stage-2 densification rates to $\dot{b}$.

HL, Ar and LZ only use temperature and accumulation rates as input variables. Other models use additional variables hypothesised to affect densification rates. These include the temperature-history mentioned above (Morris and Wingham, 2014), firn grain size (Arthern et al., 2010), impurity content (Freitag et al., 2013) and a transition region between stage-1 and

stage-2 densification (Morris, 2018). Other models are explicitly based on micro-scale deformation mechanisms (Alley, 1987; Arthern and Wingham, 1998; Arnaud et al., 2000). These efforts undoubtedly contribute to progressing towards physically based models. A potential problem with such approaches is overfitting calibration data by adding parameters to model formulations while detailed firn data remain scarce. As long as more firn data is not available to appropriately constrain the role of each variable in model formulations, we favour the use of parsimonious models relying on few input variables. It is

noteworthy that MAP$_{LZ}$, which relies on eight free parameters, performs worse on the evaluation data than MAP$_{HL}$ and MAP$_{Ar}$ with two fewer free parameters. This highlights that gains in model accuracy should rely not only on better calibration of parameters but also on a reconsideration of the governing densification equations. Additionally, firn core data invokes the assumption of a steady-state depth-density profile. As such, parameter calibration poorly captures seasonal climatic effects on densification. Comprehensive datasets of depth-density profiles (Koenig and Montgomery, 2019) are very valuable to model

development. Efforts in collecting and publishing strain rate measurements from the field (Hawley and Waddington, 2011; Medley et al., 2015; Morris et al., 2017), and possibly from laboratory experiments (Schleef and Löwe, 2013), can further benefit model calibration and the progress towards more representative equations.

In order to quantify the consequences of our calibration, we investigate two aspects for which firn models are of common use:

calculating firn compaction rates and predicting the age of firn at $z_{pc}$ depth, $age_{pc}$ (yr). At every site $i$ of our dataset, we compute the 2000-2017 total compaction anomaly, $cmp_{an,i}$ (m), and the $age_{pc,i}$ value with each of the 500 parameter sets randomly drawn from the posterior ensembles of the three different models (HL, Ar, LZ). This allows evaluation of both parameter-related and model-related uncertainty. Total compaction anomaly ($cmp_{an}$) – calculated as the cumulative anomaly in surface elevation change due only to firn compaction changes during the 2000-2017 period with respect to the climatic

reference period – is given by:

$$cmp_{an,i} = cmp_{tot,i}^{00-17} - 17cmp_{ref,i}^{yr} \tag{11}$$

where $cmp_{tot}^{00-17}$ (m) is the total firn compaction over 2000-2017 and $cmp_{ref}^{yr}$ (m yr[-1]) is the annual mean compaction over the reference period (see Sect. 2.2). At all sites, we compute the coefficients of variation (CV) for both $cmp_{an}$ and $age_{pc}$ from

the 500 simulations with each model, and we average the CVs across all sites. CV is the ratio of the standard deviation to the mean and provides an effective assessment of relative dispersion of model results. Because low mean values of $cmp_{an}$ can inflate its CV, we consider only half of the sites at which the mean computed $cmp_{an}$ is highest. For all three models, the CV values for both $cmp_{an}$ and $age_{pc}$ lie between 5.5 and 7.5% (Table 3). These values give typical uncertainty in firn model

output related to uncertain parameter values. Proceeding to the same calculations but using all three models, i.e. an inter-model ensemble of 1500 simulations at each site, gives an overview of the combined parameter- and model-related uncertainty. The CVs are 19.5% for $cmp_{an}$ and 7.5% for $age_{pc}$, demonstrating larger inter-model disagreement on $cmp_{an}$ calculations (Table 3). By using the CV values, we can calculate reasonable uncertainty estimates for $cmp_{an}$ and $age_{pc}$. For instance, in the dry snow zone of GrIS, simulated compaction anomalies are typically around 20 cm over 2000-2017, and thus come with an

uncertainty of the order of ±4 cm. Since pore close-off age here is around 250 years, a reasonable uncertainty range on this value is ±19 years. In contrast, on the drier AIS, pore close-off age is about 1000 years thus this range increases to ±75 years. Compaction anomalies hover around 0 cm on most of the dry zone of the AIS because it has not experienced the strong recent surface warming of the GrIS. Absolute uncertainty is thus reduced but still critical given the large area of the AIS over which uncertainties are aggregated when mass balance trends are evaluated. The uncertainty ranges calculated from the CV values

provide an order of magnitude of errors in firn model outputs that must be accounted for in altimetry-based mass balance assessments and in ice core studies, respectively.

We further investigate how using different models and different parameterisations leads to discrepancies in the modelled compaction. We compute monthly values of compaction anomalies over the 2000-2017 period with the original and MAP models of HL, Ar and LZ (Fig. 7). Ar shows the strongest sensitivity to climatic conditions diverging from these of the

reference period; compaction responds strongly to the general increases on GrIS in temperature and accumulation rate, especially in late summer. Due to its lower values for $\alpha, \beta$ and $E_g$, MAP$_{Ar}$ exhibits less extreme compaction anomalies than the original Ar and thus less seasonal variability. In sharp contrast to Ar, HL-computed compaction rates remain relatively stable, due to low activation energy values that smooth out the seasonal variability. Firn core observations provide little information and constraints on seasonal patterns of densification. However, it is noteworthy that MAP$_{Ar}$ and MAP$_{LZ}$ tend to

show comparable short-timescale sensitivities (insets in Fig. 7), despite structural differences in the models' governing equations. This might indicate that these models fare relatively well in capturing seasonal fluctuations of densification rates and their sensitivity to climate shifts.

**5 Conclusion**

We have implemented a Bayesian calibration method to estimate optimal parameter combinations applicable to GrIS and AIS

firn for three benchmark firn densification models (HL, Ar, LZ). An extensive dataset of 91 firn cores was separated into calibration and independent evaluation data. Two optimised models (MAP$_{HL}$, MAP$_{Ar}$) showed significant improvement against the evaluation data, while MAP$_{LZ}$ reached results close to, but slightly worse, than its original version and inferior to MAP$_{HL}$

and MAP$_{Ar}$. When compared to other models of greater complexity, the MAP models showed comparable or even improved performances. Furthermore, the Bayesian approach provides a robust way to evaluate the uncertainty related to parameter value choice, which is a major deficiency of current models. By introducing realistic climatic perturbations in the calibration process, the uncertainty intervals obtained account for the effects of an uncertain climatic forcing. However, at most sites

where we evaluated, all three models' uncertainty intervals do not cover observed *DIP* values. As such, although model results can be improved by re-calibration methods, model tuning alone is insufficient to reach exact fidelity of firn densification models. The formulation of models' governing equations impacts the remaining errors with respect to observations, which highlights deficiencies in our understanding of dry firn densification. Developing a well-constrained physically detailed model is challenging given the number of mechanisms affecting densification rates and their dependency on microstructural

properties of firn, which are difficult to observe. Our study demonstrates that, despite these observational limitations, thorough calibration methods relying only on climatic variables can substantially improve firn model accuracy, and constrain uncertainties.

*Author contributions.* VV, AL and CN conceived this study. VV performed the development of the calibration method,

performed the model experiments and led writing of the manuscript. AL and CN supervised the work. MS developed the Community Firn Model. PKM provided firn core data. BN and JMVW provided the RACMO2 forcing data. All authors provided comments and suggested edits to the manuscript.

*Data availability.* 41 of the 91 firn cores are from the SUMup dataset (2019 release), which is publicly available from the

Arctic Data Center (doi: 10.18739/A26D5PB2S). 41 of the 91 firn cores are from the dataset compiled by Matt Spencer (Spencer et al., 2001), which is available upon request. 5 of the 91 firn cores were provided by Joe McConnell and Ellen Mosley-Thompson and are available on request through PKM. 2 of the 91 cores are available via the PANGEA website (https://doi.pangaea.de/10.1594/PANGAEA.227732) and (https://doi.pangaea.de/10.1594/PANGAEA.615238). 1 of the 91 cores is available via the NOAA website (ftp://ftp.ncdc.noaa.gov/pub/data/paleo/icecore/antarctica/newall/). 1 of the 91 cores

is available via the USAP website (http://www.usap-dc.org/view/dataset/609215). All Antarctic RACMO2.3p2 climate data used are available on request through JMVW and yearly climate variables are free to download from the IMAU website (https://www.projects.science.uu.nl/iceclimate/publications/data/2018/index.php). All Greenland RACMO2.3p2 climate data used are available on request through BN and yearly SMB and components are free to download (https://doi.pangaea.de/10.1594/PANGAEA.904428). The Community Firn Model is available for download on GitHub

(https://github.com/UWGlaciology/CommunityFirnModel).

*Acknowledgements.* We thank Lora Koenig and Lynn Montgomery for making the SUMup dataset of firn cores available and easily accessible (Koenig and Montgomery, 2019). Matt Spencer is also acknowledged for publishing a separate dataset of firn cores (Spencer et al., 2001). We thank Joe McConnell and Ellen Mosley-Thompson, supported by the NSF-NASA PARCA

Project, for providing additional firn core data (Bales et al., 2009; Banta & McConnell, 2007; McConnell et al. 2002; Mcconnell et al., 2000; Mosley-Thompson et al., 2001). We thank Malcolm McMillan for his interest in the study and for providing insight into the subject of ice sheet mass balance assessments. VV thanks Elizabeth Morris for pointing out errors in geographical coordinates of some of the firn cores and for her endless interest in firn densification. The Centre for Polar

5    Observation and Modelling is acknowledged for supporting VV in his research. AL and CN research is supported by EPSRC, *A Data Science for the Natural Environment*, EP/R01860X/1. PKM acknowledges support from NESSC (Netherlands Earth System Science Centre). BN was funded by the NWO (Netherlands Organisation for Scientific Research) VENI grant VI.Veni.192.019. We thank all contributors to the development of the CFM who are not authors of this study. All authors thank two anonymous referees for their time and effort in reviewing the manuscript.

*Competing interests.* The authors declare that they have no conflict of interest.

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

| Parameter | Value in original model | Prior distribution | MAP | 95 % Credible interval |
|---|---|---|---|---|
| $k_0^*$ [m w.e.$^{-a}$] | 11 | $N(11, 100)$ | 17.4 | 7.58; 28.4 |
| $k_1^*$ [m w.e.$^{-b}$] | 575 | $N(575, 9 \times 10^4)$ | 524 | 260; 1060 |
| $E_0$ [J mol$^{-1}$] | 10 160 | $N(10160, 4 \times 10^6)$ | 10 840 | 9 000; 12 290 |
| $E_1$ [J mol$^{-1}$] | 21 400 | $N(21400, 4 \times 10^6)$ | 20 800 | 18 900; 22 300 |
| $a$ [/] | 1 | $N(1, 0.4)$ | 0.91 | 0.74; 1.02 |
| $b$ [/] | 0.5 | $N(0.5, 0.4)$ | 0.63 | 0.54; 0.78 |
| $k_0^{Ar}$ [m w.e.$^{-\alpha}$] | 0.07 | $N(0.07, 4.9 \times 10^{-3})$ | 0.077 | 0.046; 0.137 |
| $k_1^{Ar}$ [m w.e.$^{-\beta}$] | 0.03 | $N(0.03, 9 \times 10^{-4})$ | 0.025 | 0.015; 0.048 |
| $E_c$ [J mol$^{-1}$] | 60 000 | Fixed: 60000 | / | / |
| $E_g$ [J mol$^{-1}$] | 42 400 | $N(42400, 16 \times 10^6)$ | 40 900 | 39 700; 42 000 |
| $\alpha$ [/] | 1 | $N(1, 0.4)$ | 0.80 | 0.66; 0.89 |
| $\beta$ [/] | 1 | $N(1, 0.4)$ | 0.68 | 0.59; 0.81 |
| $lz_a$ | 8.36 | $N(8.36, 36)$ | 7.31 | 3.93; 12.82 |
| $lz_b$ | -2.061 | $N(-2.061, 2)$ | -2.124 | -2.319; -1.896 |
| $lz_{11}$ | -9.788 | $N(-9.788, 36)$ | -14.710 | -20.839; -5.469 |
| $lz_{12}$ | 8.996 | $N(8.996, 36)$ | 7.269 | 2.680; 17.724 |
| $lz_{13}$ | -0.6165 | $N(-0.6165, 1)$ | -1.019 | -1.389; -0.509 |
| $lz_{21}$ | -2.0178 | $N(-2.0178, 2)$ | -1.513 | -2.970; -0.258 |
| $lz_{22}$ | 8.4043 | $N(8.4043, 36)$ | 6.0203 | 4.911; 12.942 |
| $lz_{23}$ | -0.0932 | $N(-0.0932, 0.25)$ | -0.0913 | -0.133; -0.0460 |

**Table 1.** Information for the free parameters of HL (top), Ar (middle) and LZ (low). $N(x, y)$ designates a normal distribution of mean $x$ and variance $y$. The variances in the prior distributions are taken to generate weakly informative distributions. Some prior correlation is prescribed for the pairs $(k_0^*, E_0)$, $(k_1^*, E_1)$, $(k_0^{Ar}, E_g)$, $(k_1^{Ar}, E_g)$ and $(k_0^{Ar}, k_1^{Ar})$ (see Supplementary Information). MAP estimates and credible intervals are results from the calibration process.

| Model | RMSE ($DIP15$) [m] | RMSE ($DIPpc$) [m] |
|---|---|---|
| HL original | 0.503 | 2.395 |
| HL MAP | 0.382 | 1.862 |
| HL 500 random sample | 0.396 | 1.899 |
| Ar original | 0.772 | 4.566 |
| Ar MAP | 0.426 | 1.780 |
| Ar 500 random sample | 0.448 | 1.889 |
| LZ original | 0.452 | 1.812 |
| LZ dual | 0.505 | 3.883 |
| LZ MAP | 0.463 | 2.392 |
| LZ 500 random sample | 0.486 | 2.296 |
| IMAU-FDM | 0.418 | 2.681 |

**Table 2.** Model results on the evaluation data. The Root Mean Squared Errors (RMSE) are calculated with respect to the observations of depth integrated porosity until 15 m depth and until pore close-off.

| Coefficient of Variation | HL | Ar | LZ | Combined (HL, Ar, LZ) |
|---|---|---|---|---|
| $cmp_{an}$ | 5.8% | 5.8% | 6.5% | 19.5% |
| $age_{pc}$ | 6.5% | 5.8% | 7.5% | 7.5% |

**Table 3.** Coefficients of variation for the 2000-2017 cumulative compaction anomaly ($cmp_{an}$) and firn age at pore close-off depth ($age_{pc}$).

5    Values are computed from results of 500 randomly selected parameter combinations from the posterior ensembles of each model (HL, Ar, LZ). Coefficients of variation are averaged across all sites of the dataset.

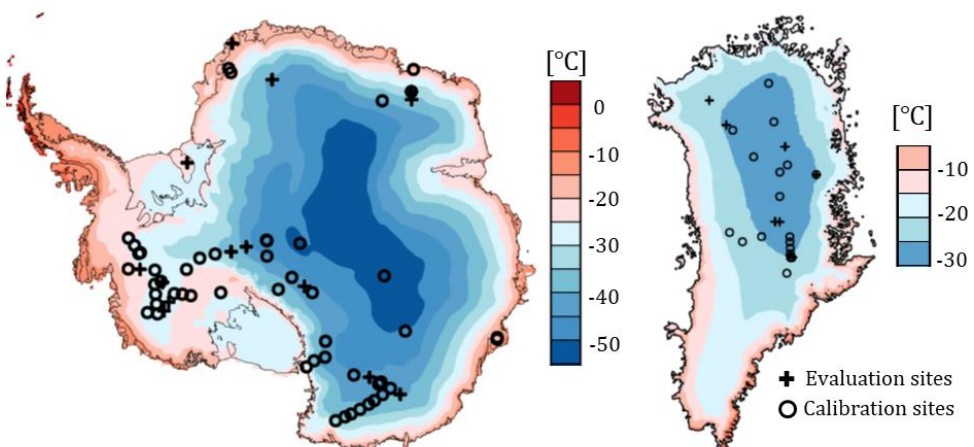

**Figure 1.** Maps of Antarctic (left) and Greenland (right) ice sheets. Background is mean annual air temperature as modelled by RACMO2.

10    Note the different colour scales.

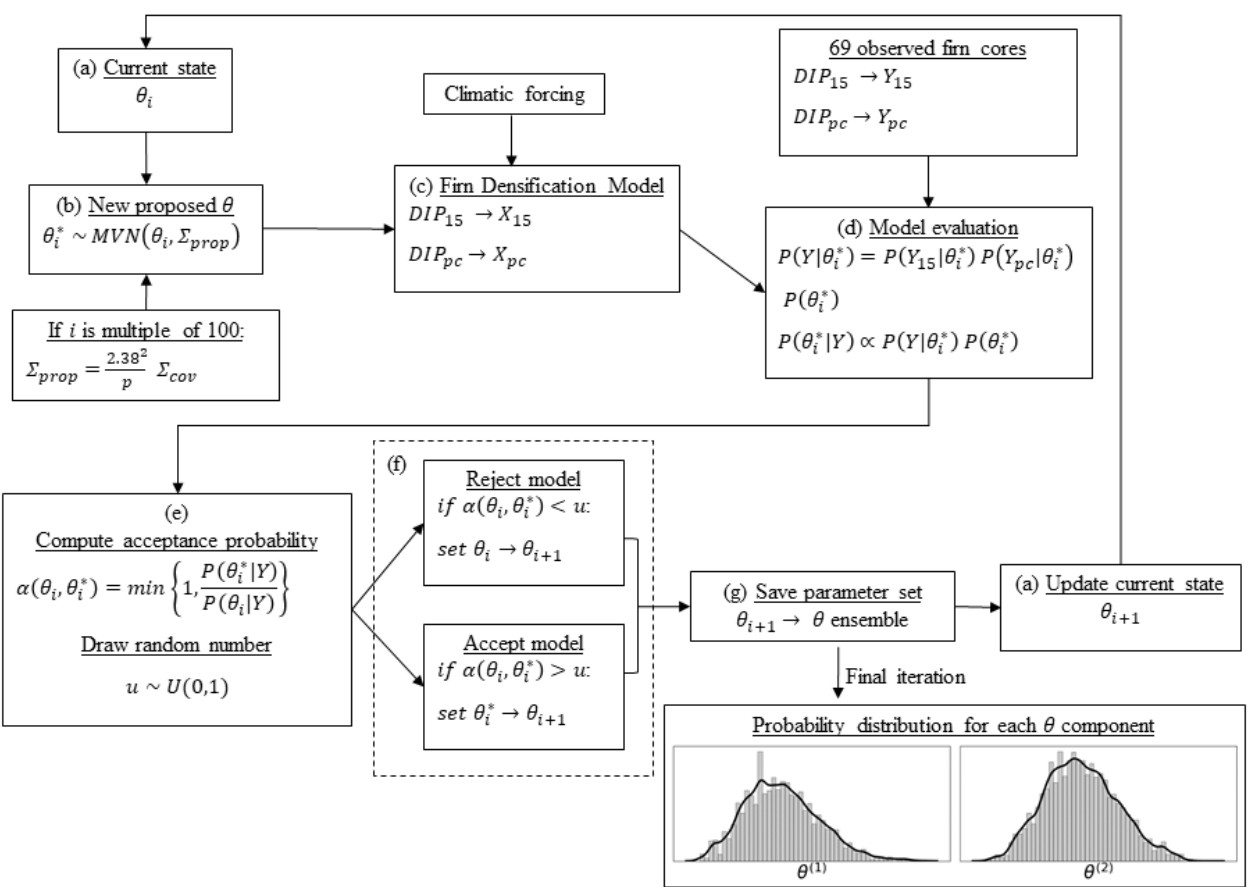

**Figure 2.** Implementation of the Random Walk Metropolis algorithm. $\theta$ represents a parameter combination of any given firn densification model investigated.

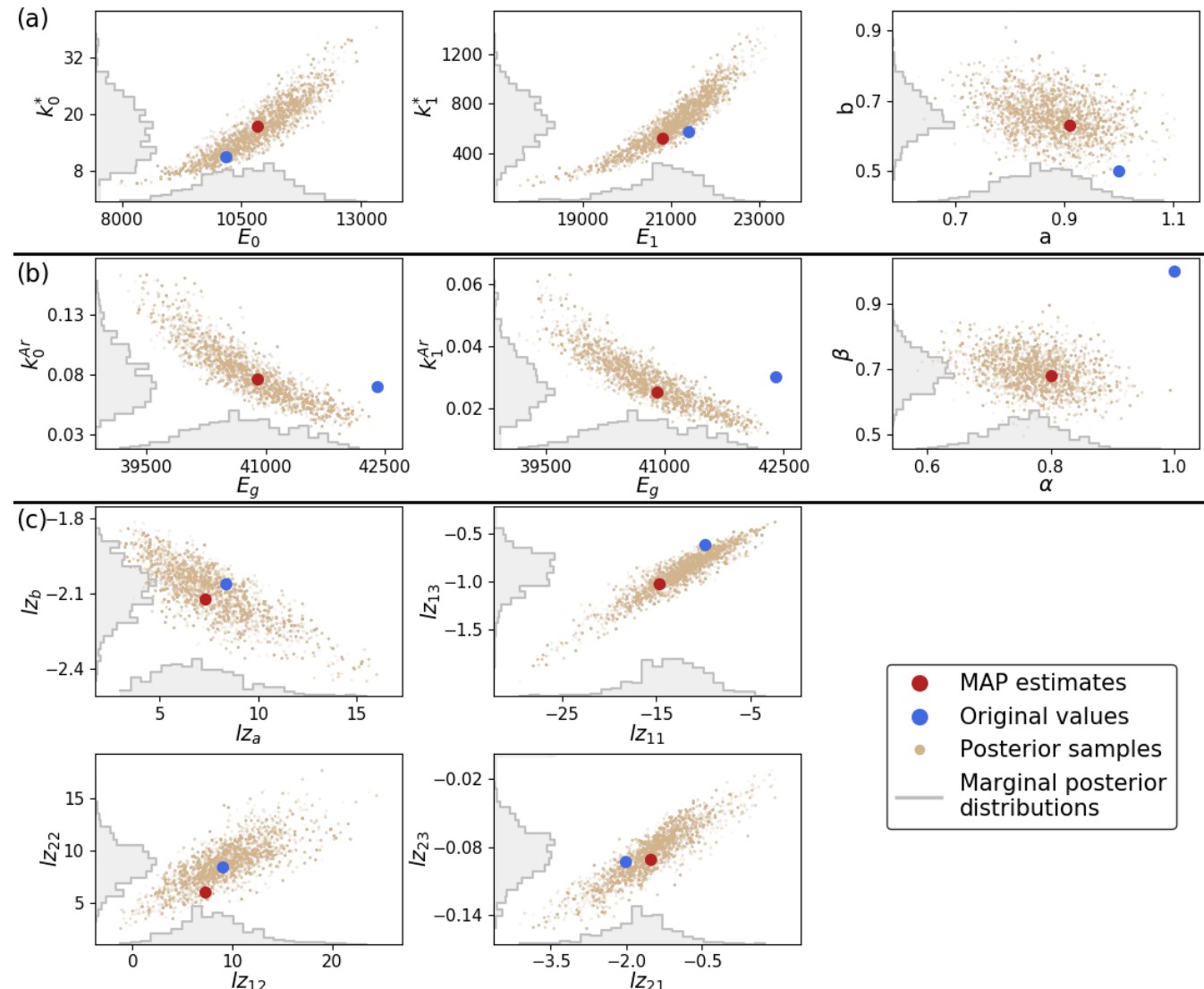

**Figure 3.** Posterior probability distributions, shown for pairs of parameters, for (a) HL, (b) Ar, (c) LZ. Where possible, correlated parameters share the same graph (see Supplementary Information for full correlation matrices). The posterior samples are 500 randomly selected parameter combinations from the posterior ensembles of each model (HL, Ar, LZ).

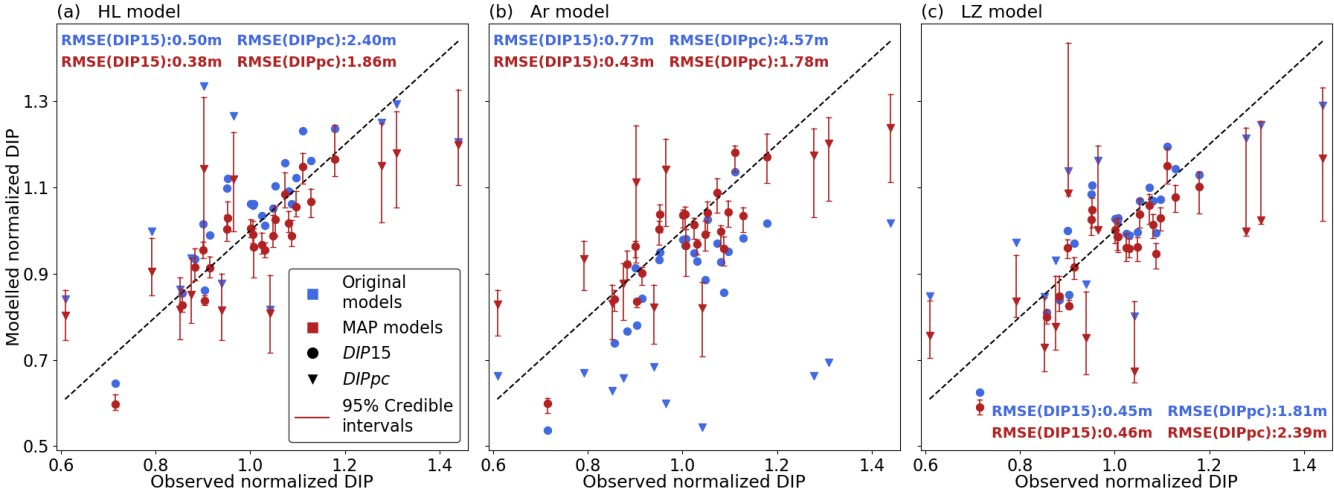

**Figure 4.** Comparison of evaluation data *DIP* with model results. The 95% credible intervals are computed from results of 500 randomly selected parameter combinations from the posterior ensembles of each model (HL, Ar, LZ). Similar scatter plots for the LZ dual and IMAU results are shown in the Supplementary Information (Fig. S6).

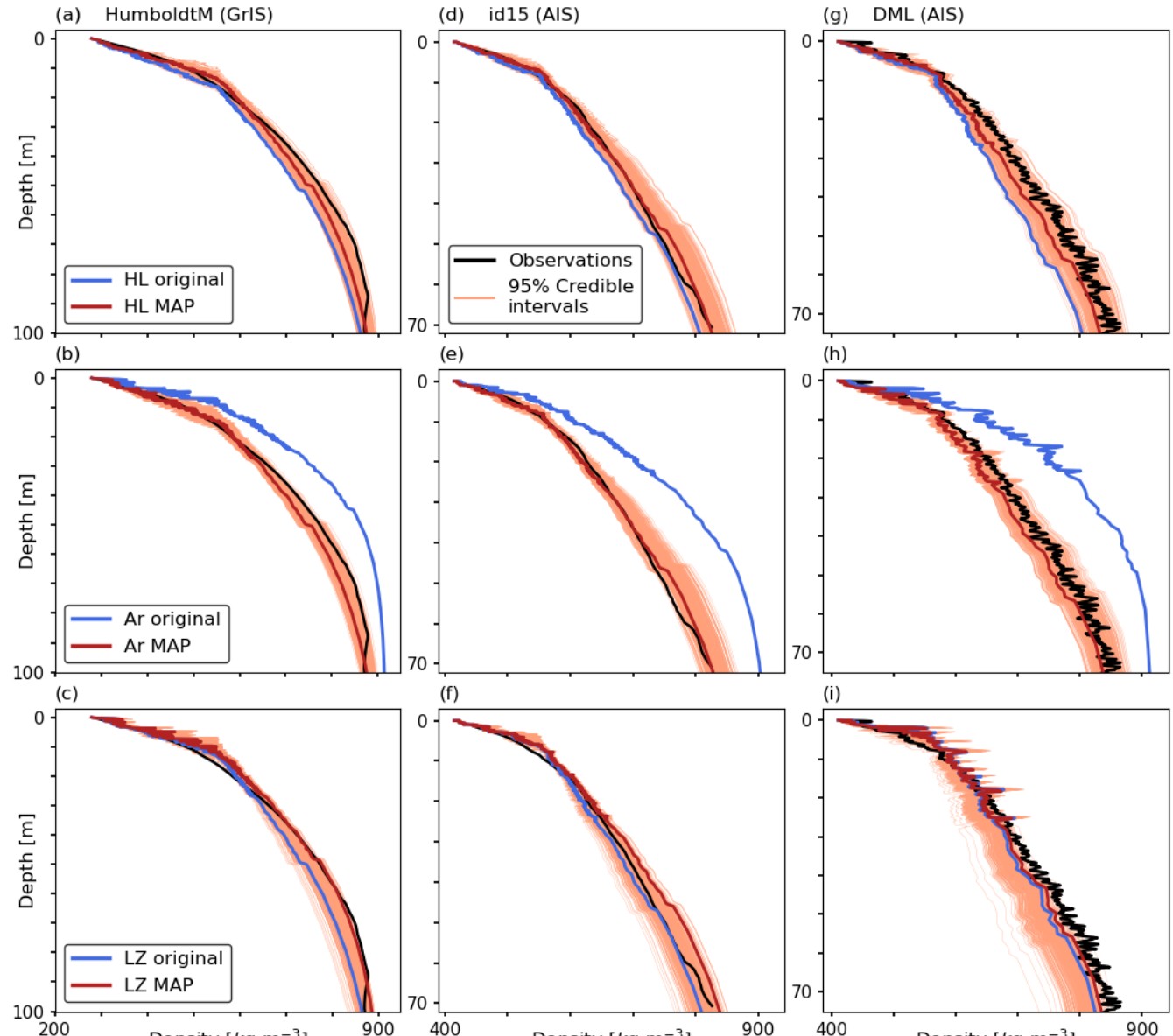

**Figure 5.** Depth-density profiles at three evaluation sites. DML is a climatic outlier of our dataset with particularly high temperatures and accumulation rates. The 95% credible intervals are computed from results of 500 randomly selected parameter combinations from the posterior ensembles of each model (HL, Ar, LZ).

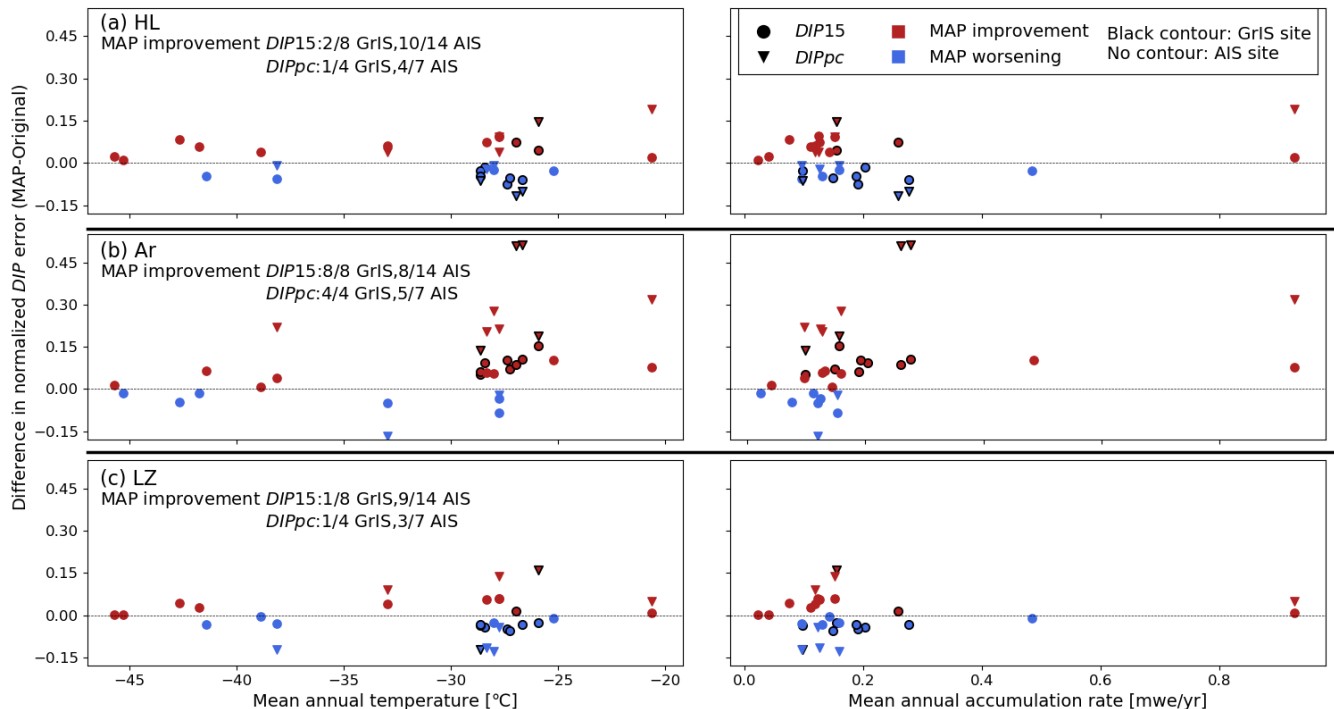

**Figure 6.** Improvements of the MAP models with respect to the original models for the evaluation data. The ratios indicate the ratios of cores for which an improvement is achieved by the corresponding MAP. Graphs in the left column display the mean annual temperature on the x-axis and those in the right column display the mean annual accumulation rate.

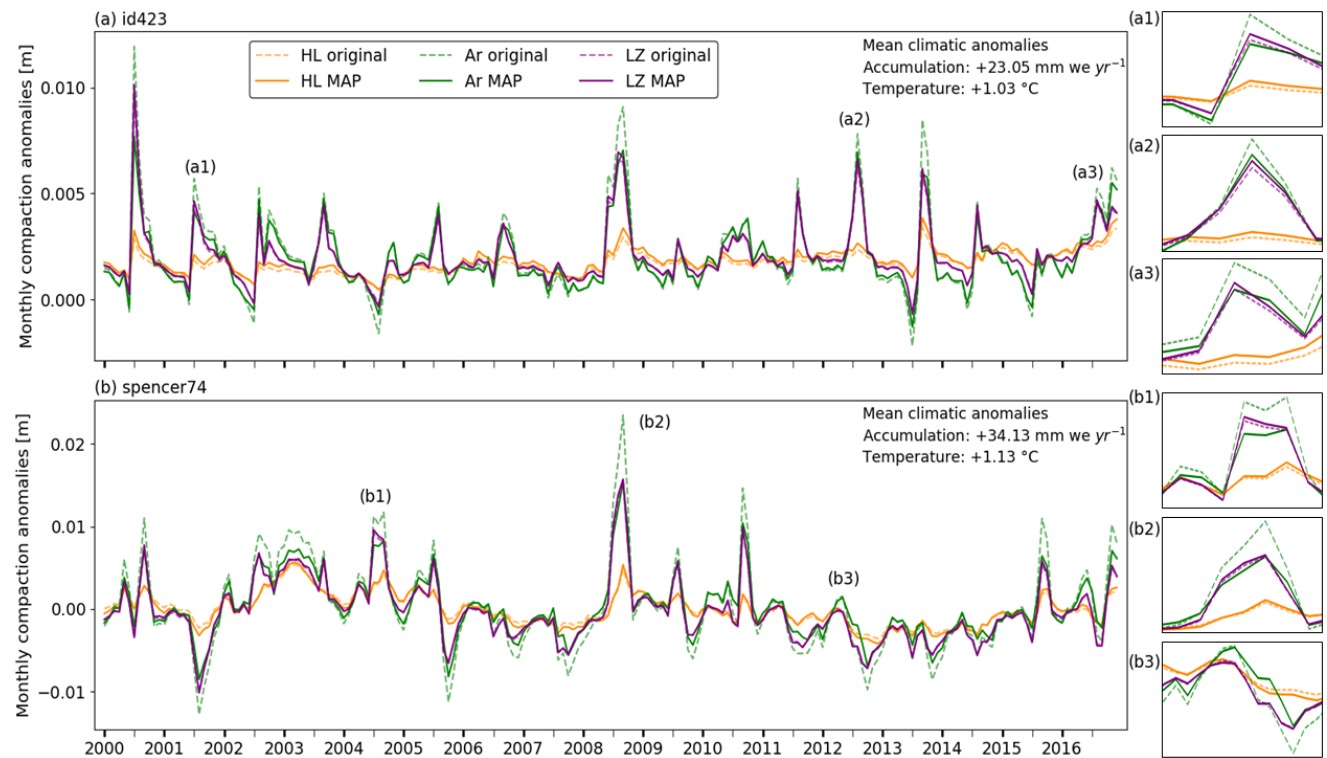

**Figure 7.** Monthly time-series of compaction anomalies at two sites on the GrIS. Insets show details for particular intervals of the time-series. Mean climatic anomalies are calculated as a difference between mean climatic values over the period 2000-2017 with respect to the reference period 1960-1979, and based on RACMO2 values.

