# Peer review of "Bayesian calibration of firn densification models"

_The Cryosphere, 2019_

## Referee Comment (RC1) · Anonymous Referee #1 · 2 Mar 2020

The study by Verjans et al. focuses on Bayesian calibration of parameters in three widely used firn models. All three models simulate dry firn compaction; they do not consider the influence of meltwater percolation and ice layer formation in firn. The authors use a comprehensive set of firn cores from dry firn areas of Antarctica and Greenland. The majority of the cores is used for model calibration; the remainder is used for model validation. The authors find that the Bayesian calibration results in model parameters that differ from the original parameters. Validation of original and newly calibrated models shows that the Bayesian calibration improved performance of one model substantially, one model was slightly improved and the third model showed no improvement.

The study by Verjans et al. appears sound and concise to me. The presentation of the research appears of high quality. However, my expertise in Bayesian approaches is limited and thus I did not focus my evaluation on the implementation of the Bayesian

calibration. Below I provide a list of general remarks.

- The authors point out that "Results of the calibration would depend on the particular climate model used for forcing" (Line 4, Page 4). This important point is mentioned in Data and Methods, but not in Discussion and Conclusions. From reading the latter two sections, I gained the impression that the authors suggest replacing the original parameters with the MAP parameters (with the exception of the LZ model). However, could the difference in parameter values also result from different model forcing? Which parameters would the Bayesian calibration provide as output if you would use, for example, the climate conditions assumed by Herron and Langway (1980)? I understand that this is difficult to quantify, but I suggest at least to highlight this in Discussion and Conclusions or to test the stability of the calibration under different forcing.

- I appreciate that the authors investigate the impact of the new calibration on a Greenland scale. Nevertheless, I feel the comparison could be improved by showing the numbers in the context of total Greenland mass change. Furthermore, the three models are designed to simulate dry firn compaction, while the sensitivity analysis extends to the entire accumulation area. A very substantial part of the Greenland accumulation area is subject to melt and refreezing. How does this influence the informative value of your sensitivity analysis?

- Figures: I appreciate the good quality of the figures. My only suggestion is to use the same colour scale for Greenland and Antarctica in Figure 1. As it is now, and being fully aware that the two colour bars are different, it is difficult to anticipate the differences in climate at the core locations. For both maps, why does the colour bar represent a temperature range that exceed the actual range in climate conditions?

- References: I had only a brief look at the references, but noticed that the bibliography for Shepherd et al., Science, 2012, might contain some errors. Looking

up the article, I found for example a different DOI ("10.1126/science.1228102" instead of "5b0143 [pii]").

---

## Referee Comment (RC2) · Anonymous Referee #2 · 21 Mar 2020

The paper aims to evaluate and improve the parametrisations of 3 firn densification models, for the dry snow part. It is written in a concise and clear matter, but I find that the scope of the paper is too narrow. The authors use a sound method, and find weakly different results from the original publications for 2 of the 3 models. They make a good effort in discussing the implications of their findings in terms of densification physics, but still, to my mind, they consider only part of the problem, and it is difficult to make use of their findings without the rest of the picture.

For instance, in the conclusion, they say "As such, although model results can be improved by re-calibration methods, model tuning alone is insufficient to reach exact fidelity of firn densification models. The formulation of models' governing equations impacts the remaining errors with respect to observations, which highlights deficiencies in our understanding of dry firn densification." The problem is that they don't evaluate all parts of their models.

[Figure]

I consider that the parts of the (inverse) problem are : 1. Input data. 1.1 Away from meteorological stations, surface temperature and accumulation are not well known. Even if RACMO is as good as it gets, there are biases that can be several °C in temperature. Simply assuming that RACMO is right is not acceptable. A range of scenarios, based on the known biases of RACMO, or uncertainty derived from also using MAR, and different reanalyses is necessary. In many cases, you can find a mean accumulation derived from the ice core the density was measured on, why not use that? 1.2 surface density. I support the idea to use measured density, but measuring surface density is not easy. It comes with an uncertainty of 20% at least. You need to do a sensitivity study of your whole process with randomy different surface density within a reasonable uncertainty range. Surface density is actually a pretty sensitive parameter.

2. Assumptions that go into the model, like the steady state assumption, and the 1D assumption. In some areas, we know these assumptions are not right. In your outliers to the models, when it is an outlier to all models, did you consider that maybe some of these assumptions were not right? Mizuo is for instance a very strange site. Places with snow dunes in Antarctica can do funny things too..

3. the physical formulation : you discussed this thing well, but did not combine all your model output to give the global uncertainty, or compare the within-model to the across-models uncertainty. For instance, it would be interesting to see a fouth row on figure 5 with the outputs of the 3 models on the same figure, and comment why the uncertainty ranges don't overlap. I am surprised by that, and it makes me think that your 95% confidence interval is underestimated.

4. The parametrisation of each model. You focused on this. My major problem with the method you used is that you assumed that your parameters were independent, when in reality they are not at all, as shown the covariance matrix on Fig S4 (e.g. k1 and E1 having a covariance of 0.94). You don't show your prior error covariance matrix, so it's difficult to assess exactly what you did. A better description in the prior assumptions, including a comparison of the prior and posterior probability distributions (for instance

adding the prior to Fig 3) is needed. From what I read, you included no covariance in your parameters in your prior, and that is not right. For instance, in fig 3, showing k1* and E1 independently makes no sense, because you could always compensate any error in E1 by a change in k1*, it would make more sense to show these either in a 2D plot, or for a fixed E1 in the case of k1 and vice versa.

5. The metric used to compare models to data, and the quality of the data. 5.1 Data. Here, I'm with you, I don't want to go through checking each dataset again, but you should at least give some information about these data, citing the spencer paper, and maybe a few other, to illustrate what is the uncertainty in the data , given different methods (weighting a full core is not quite the same as doing gamma, or CT..). 5.2 Metric. Is DIP really the best metric? Is it faithful? You are the first to use this metric for the calibration of firn air models. Everyone else was using rho(z) directly. Does it give the same answer? You should demonstrate that. Is it a good metric also for other applications of firn modelign, such as the close-off depth estimation? A paragraph demonstrating the usefullness and validity of this metric would be nice.

The general objective of the paper is, if I quote the abstract to "demonstrate how model- and parameter-related uncertainties potentially affect ice sheet mass balance assessments". It's a great idea, and such a thorough assessment of firn models has not been done. Lundin et al. 2017 highlighted some model deficiencies, but did nothing to remedy those. I support a future version of this paper to be published with a quantitative answer to this question, but we are not there yet.

After you have properly addressed all the parts of the problem stated above, I'd like to see a quantitative comparison of the different sources of uncertainty. If I dont want to run the community firn model (CFM) 30,000 times but just a few, to get a gist of the uncertainty in my specific application, should I use a wide range of (T, accum) scenarios (step 1)? or rather use different model physics (step 3)? Or one of the models, but with a range of parameters (step 4)? This is a practical question that would be really useful to future users of the community firn model. And finally, when

you have done that, it would be great to go back and recalculate the uncertainty in mass balance from altimetry, using the above mentionned decomposition. This is the great paper I'd like to read. It's marginally more work from what you have done, running a few more simulations on the same framework, but I think it would be really worth it.

I would like to finish with more detailed comments: 1. Method: Why did you go for Monte-Carlo, rather than a form of generalized least squares, which would have converged in <10 runs very likely, and could have easily dealt with covariance in model parameters? I agree that this problem is non linear, and underdetermined, but it is also monotonic, so least squares are applicable, and converge much faster. That being said, your method is valid.

2. choice of models. Why did you choose the Arthern model as one of your 2 models, rather than the IMAU versions (ligtenberg or kuipers)? We already know that the physical formulation of Arthern is not right (Lundin 2017). You later discuss the IMAU version. I think it would make more sense to publish an optimisation for these rather than the Arthern, which shows no sensitivity to the accumulation rate, something we know is wrong.

---

## Author Comment (AC1) · 17 Apr 2020

The authors would like to thank the reviewer for their comments. We are currently working on revising the manuscript in order to address their remarks. The main comment about sensitivity of the calibration to the climatic forcing is sensible and we focus strongly on this point in our revision. We will address the shortcomings of the Discussion and Conclusion sections that the reviewer pointed out. We hope to complete the revisions as soon as possible and we will provide a more in-depth response to the reviewer once the manuscript will have been updated. Thank you very much for taking the time to review our study.

---

## Author Comment (AC2) · 17 Apr 2020

The authors would like to thank the reviewer for their comments. We are currently working on revising the manuscript in order to address their remarks. The reviewer has pointed out numerous issues that need to be addressed and/or clarified in the paper. The question of sensitivity to uncertainty in boundary conditions is important and we focus strongly on this aspect in the revision. We also plan to justify our performance metric better, to modify our prior assumptions in the Bayesian framework and to improve the interpretability of our results. Furthermore, we will clarify the scope of our study. These changes require new simulation experiments and thus a re-analysis of the results. We hope to update the current manuscript as soon as possible. Along with the submission of the revised manuscript, we will give a more detailed response to all the reviewer's comments. Thank you very much for taking the time to review our study.

---

## Author Response (AR1)

Dear Dr. Whitehouse,

We would like to thank you and both reviewers for your attention to our manuscript. We appreciate the interest of the reviewers in our study, their constructive comments and their help in improving the quality of the paper. We have provided detailed individual responses to each reviewer and we first highlight here the major modifications in the revised manuscript with respect to the first version:

**The definition of our research theme**

We clarify that this study is focused on the methodology of implementing a Bayesian calibration method for firn models. Furthermore, we aim at demonstrating how our findings of uncertainty related to firn model parameterisation affect firn model output. In this way, we try to show more specifically the direct impact of our results for the broader community of firn model users. We also exploit more the Bayesian framework of our study in order to illustrate how this method can be of practical use to other modellers of the glaciology community. We emphasize that this study is not an estimation of uncertainty in ice sheet wide mass balance estimates from altimetry.

For these purposes, we made the following modifications:

a) We added a section discussing both intra- and inter-model spread on firn model output. The spread in results are computed by exploiting posterior ensembles of parameters. The firn model outputs investigated are compaction rates and the age of firn at the firn-ice transition because both these variables are of important interest to firn model users.

b) We removed the paragraph approximating the impact of our results in terms of compaction rates if they were to be upscaled at the scale of the Greenland ice sheet. These calculations were too speculative, and a thorough uncertainty analysis would require taking many other aspects into account, which is beyond the scope of this study. We subsequently modified the abstract to remove any mention of ice sheet wide estimations.

**The boundary forcing**

As highlighted by the reviewers, it was important to account for uncertain forcing of firn models: the climatic input and the surface density conditions. In this revised version of our study, we allow realistic magnitudes of these uncertainties to propagate into the model calibration process and thus, to influence the parameter estimates and their credible intervals. This was implemented by adding random perturbations in these fields. The climatic random perturbations are based on typical values of RACMO2 errors and biases with respect to weather station measurements. We provide all the details of the implementation of the random perturbations in the updated manuscript and its Supplementary Information. We note however that this study is not a complete sensitivity analysis to climatic conditions and/or to fresh snow density. Our goal here is to let reasonable estimates of errors in those fields to be accounted for in the calibration process.

**Interpretability of our results**

As mentioned above, we discuss more thoroughly intra- and inter-model spread for variables of interest to the firn science community. We believe that the coefficients of variation, expressed in %, provide a straightforward overview to the reader of the uncertainty one can expect in firn model output. In general, we focus more on the implications of our findings in terms of uncertainty than on the specific comparison between performances of the MAP and original models. We also facilitated the understanding of the multi-dimensionality of the calibration method for the reader. As suggested by Reviewer 2, Figure 3 has been changed to explicitly demonstrate correlations between different parameters.

Many additional modifications have been integrated to the manuscript and all of these are detailed and explained in our responses. Responses to the reviewers and a marked-up version of the updated manuscript are provided below.

Best regards,
Vincent Verjans, on behalf of authors

**Response to Reviewer 1**

We thank the reviewer for their effort in reading and evaluating our study. Their comments were accounted for and we believe that our subsequent modifications have improved our study. Our response to their comments is provided below. The original text from the referee is in black italic and our responses are in blue. Throughout the response, we refer the reviewer to the revised manuscript for evaluating the modifications to the study. A marked-up copy of the revised manuscript is attached below (p.9 and below).

*The authors point out that "Results of the calibration would depend on the particular climate model used for forcing" (Line 4, Page 4). This important point is mentioned in Data and Methods, but not in Discussion and Conclusions. From reading the latter two sections, I gained the impression that the authors suggest replacing the original parameters with the MAP parameters (with the exception of the LZ model). However, could the difference in parameter values also result from different model forcing? Which parameters would the Bayesian calibration provide as output if you would use, for example, the climate conditions assumed by Herron and Langway (1980)? I understand that this is difficult to quantify, but I suggest at least to highlight this in Discussion and Conclusions or to test the stability of the calibration under different forcing.*

The close link between firn model parameters and climatic forcing is a challenge for firn modelling and the dependence of the parameterisation to the forcing is unavoidable. In order to try and account for this, we introduce uncertainty in the climatic forcing in the calibration process by applying random perturbations to both the temperature and accumulation fields of RACMO2. These perturbations are based on published errors and biases of RACMO2 with respect to weather station data. We explain this in Section 2.2 (p.11 l.24) and provide all the technical details of the implementation in the Supplementary Information (Section S2). We hope that this addresses the issue of *"stability of the calibration"*. This development leads to larger uncertainty ranges of the parameters, as we expected. The intervals now incorporate ranges that one can expect to reach using any realistic climatic forcing. We believe that this demonstrates even further the benefits of the Bayesian approach: the interesting outcome of a calibration is not only the best-fit parameter set (i.e. the MAP), but also the robust uncertainty ranges. Accounting for this uncertainty is crucial for any firn model application. As suggested by the reviewer, we add an extra sentence in the Conclusion (p.18 l.5) to raise the awareness of the readers to this issue. Please note that we introduce random perturbations in surface density also, since this is another uncertain boundary condition for firn densification models.

*I appreciate that the authors investigate the impact of the new calibration on a Greenland scale. Nevertheless, I feel the comparison could be improved by showing the numbers in the context of total Greenland mass change. Furthermore, the three models are designed to simulate dry firn compaction, while the sensitivity analysis extends to the entire accumulation area. A very substantial part of the Greenland accumulation area is subject to melt and refreezing. How does this influence the informative value of your sensitivity analysis?*

In hindsight, we agree with the reviewer that this section was a little speculative. A thorough sensitivity analysis would require taking many other aspects into account, including the *"melt and refreezing"*. Such ice sheet scale sensitivity analysis is beyond the scope of our study and we have removed this section of the Discussion. Instead, we show in a simple example how the Bayesian uncertainty assessment translates into uncertainty in model outputs (p.16 l.49). We use two metrics that are of interest to the broad firn science community: the compaction rate and the age of firn at pore close-off depth. This replacement renders our Discussion section less speculative, more embedded in the Bayesian framework and clearer about the implications of our findings for the wider glaciology community.

*Figures: I appreciate the good quality of the figures. My only suggestion is to use the same colour scale for Greenland and Antarctica in Figure 1. As it is now, and being fully aware that the two colour bars are different, it is difficult to anticipate the differences in climate at the core locations. For both maps, why does the colour bar represent a temperature range that exceed the actual range in climate conditions?*

We thank the review for his comment on the quality of our figures. Following their suggestions, we have tried many possible alternatives for Figure 1. Using the same colour scale for both ice sheets resulted in a very red-looking Greenland. Thus, we tried using only a blue colour scale to avoid this problem. The result was not satisfactory (see the

figure at the end of the document). We have thus decided to keep the original map. However, the colour bars are modified as the reviewer suggested, and we have added a statement "Note the different colour bars" in the caption.

*References: I had only a brief look at the references, but noticed that the bibliography for Shepherd et al., Science, 2012, might contain some errors. Looking up the article, I found for example a different DOI ("10.1126/science.1228102" instead of "5b0143 [pii]").*

We reviewed our list of references. Please note that the citation to the study of Shepherd et al. (2012) has been replaced by a more recent altimetry-based assessment of Antarctic mass balance: Shepherd et al. (2019).

[Figure]

**Response to Reviewer 2**

We thank the reviewer for the comments provided and for their effort in bringing forward suggestions for improving our study. Based on their comments, we proceeded to several adjustments and we believe that these improve the quality and robustness of our methodology. Our response to their comments is provided below. The original text from the referee is in black italic and our responses are in blue. Throughout the response, we refer the reviewer to the revised manuscript for evaluating the modifications to the study. A marked-up copy of the revised manuscript is attached below (p.9 and below).

*The paper aims to evaluate and improve the parametrisations of 3 firn densification models, for the dry snow part. It is written in a concise and clear matter, but I find that the scope of the paper is too narrow. The authors use a sound method, and find weakly different results from the original publications for 2 of the 3 models. They make a good effort in discussing the implications of their findings in terms of densification physics, but still, to my mind, they consider only part of the problem, and it is difficult to make use of their findings without the rest of the picture.*

We would like to emphasize that this manuscript is mainly concerned with presenting an advanced methodology which we hope will contribute to a more rigorous use of models in firn science and glaciology in general. Following the advice of the reviewer, we aimed at clarifying the scope of our study in the manuscript. We have also tried to better demonstrate how our findings, and a Bayesian approach to model calibration in general, can contribute to the research of the broader glaciology community. Throughout this response, we try to demonstrate how the updated version of the manuscript fulfils these aims.

*Away from meteorological stations, surface temperature and accumulation are not well known. Even if RACMO is as good as it gets, there are biases that can be several ℃ in temperature. Simply assuming that RACMO is right is not acceptable. A range of scenarios, based on the known biases of RACMO, or uncertainty derived from also using MAR, and different reanalyses is necessary.*

We agree that the first version of the study was not sufficiently addressing the problem of the dependency of parameter calibration to climatic forcing. In order to better take this into account, we modify our calibration process. We introduce random perturbations to both the temperature and accumulation fields from RACMO2. These perturbations are based on published errors and biases of RACMO2 with respect to weather station data. We explain this in Section 2.2 (p.11 l.24) and provide all the technical details of the implementation in the Supplementary Information (Section S2). With these random perturbations, the credible intervals of our parameter values have become larger as we expected. The intervals now incorporate ranges that one can expect to reach using any realistic climatic forcing. We believe that this demonstrates even further the benefits of the Bayesian approach: the interesting outcome of a calibration is not only the best-fit parameter set (i.e. the MAP), but also the uncertainty ranges. We decided to follow the first suggestion of the reviewer and not use other forcing products such as MAR or reanalyses products. Our point of view is that any study making use of firn models should be aware of their empirical nature. Ideally, a model calibration to the specific forcing used should be performed before any application of firn models. The goal of this study is not to provide a specific "best fit" parameter combination for each one of the numerous climatic products that are commonly used (RACMO, MAR, HIRHAM, CESM, NHM-SMAP, ERA, NCEP, MERRA, etc.). Therefore, we have decided to follow the approach of introducing the observation-based random climatic perturbations. We consider that the credible intervals obtained are representative of the range of parameter values that would be applicable using any of these forcing products.

As the reviewer points out, one cannot *"assume that RACMO is right"*. But firn models are generally used coupled to climatic models, and using these climatic models for calibrating firn model parameters is thus sensible. The addition of random noise aims at making the parameterisation less model-specific. We add an extra sentence in the Conclusion (p.18 l.5) in order to clarify this.

*In many cases, you can find a mean accumulation derived from the ice core the density was measured on, why not use that?*

Our main reason for not using accumulation rates derived from the ice core is that we use a dynamic climatic forcing. The advantage of accumulation rates produced by a climate model is that we have time-varying values and not only a

long-term mean or annual values. In several applications of firn models (e.g. correction of altimetry measurements and surface mass balance modelling), it is crucial that the model outputs remain as correct as possible throughout the year. However, monthly accumulation rates can be strongly different from long-term or annual means. A second reason for our approach is mentioned in the response above: firn models are mostly used forced with climate model products. Thus, calibrating the parameters with these products is a reasonable and pragmatic approach, although climate models include unavoidable errors.

*1.2 surface density. I support the idea to use measured density, but measuring surface density is not easy. It comes with an uncertainty of 20% at least. You need to do a sensitivity study of your whole process with randomy different surface density within a reasonable uncertainty range. Surface density is actually a pretty sensitive parameter.*

We followed the advice of the reviewer to introduce random noise in the surface density boundary condition. This is also added in Section 2.2 (p.11 l.35) and explained in greater details in the Supplementary Information (section S2). Our point of view is similar to that with respect to the climatic boundary condition: the goal of this study is not to proceed to a thorough sensitivity analysis of firn model output to surface density. But the reviewer is correct that uncertainty in surface density must propagate in the calibration process and must ultimately be translated into uncertainty in firn model output. We believe that the addition of random noise fulfils this aim.

*Assumptions that go into the model, like the steady state assumption, and the 1D assumption. In some areas, we know these assumptions are not right. In your outliers to the models, when it is an outlier to all models, did you consider that maybe some of these assumptions were not right?*

We agree with the reviewer that many assumptions inherent to current firn models are not physically realistic. Our goal is to investigate the parameterisation of existing, and commonly used, firn models. We have not developed a new firn model. We hope that future developments to firn models will make them closer to the physical processes underlying firn densification. Please note that we highlight these structural shortcomings of firn models in the conclusion (p.18 l.7).

*the physical formulation : you discussed this thing well, but did not combine all your model output to give the global uncertainty, or compare the within-model to the acrossmodels uncertainty. For instance, it would be interesting to see a fouth row on figure 5 with the outputs of the 3 models on the same figure, and comment why the uncertainty ranges don't overlap. I am surprised by that, and it makes me think that your 95% confidence interval is underestimated.*

In order to better evaluate both intra-model and inter-model uncertainties, we have added an analysis of variability in firn model output in the Discussion (p.16 l.49). We focus on compaction anomalies and firn age at the pore close-off depth because these are firn model outputs of common interest. The parameter-related (i.e. intra-model) uncertainties can be directly compared with the combined parameter- and model-related (i.e. inter-model) uncertainty. This analysis is based on the standard deviations obtained from the posterior ensembles of parameter combinations and thus further exploits the benefits of our Bayesian approach. It highlights that uncertainties are larger when accounting for inter-model spread, especially for compaction rates.
Concerning Figure 5, it is noticeable that the uncertainty ranges have expanded due to the addition of climatic noise. The uncertainty ranges of the three models now clearly overlap. We think that the reader can compare the ranges of the three models because the observation profile, the x-axis and the y-axis are the same for the three models at a given site.

*My major problem with the method you used is that you assumed that your parameters were independent, when in reality they are not at all, as shown the covariance matrix on Fig S4 (e.g. k1 and E1 having a covariance of 0.94). You don't show your prior error covariance matrix, so it's difficult to assess exactly what you did. A better description in the prior assumptions, including a comparison of the prior and posterior probability distributions (for instance adding the prior to Fig 3) is needed. From what I read, you included no covariance in your parameters in your prior, and that is not right. For instance, in fig 3, showing k1\* and E1 independently makes no sense, because you could always compensate any error in E1 by a change in k1\*, it would make more sense to show these either in a 2D plot, or for a fixed E1 in the case of k1 and vice versa.*

Following the comments of the reviewer, we have modified the prior distributions of pairs of parameters for which we have an a priori knowledge of some correlation structure. The pairs of parameters are $(k_0^*, E_0)$, $(k_1^*, E_1)$, $(k_0^{Ar}, E_g)$ and $(k_1^{Ar}, E_g)$. We provide this information in Section 2.4 (p.12 l.40) and in the caption of Table 1. We provide all the technical details of the estimation of the prior correlations in the Supplementary Information (section S3).

5  It is important to keep in mind that prior independence does not imply posterior independence. If any correlations are plausible given the data, posterior distributions can show, potentially very strong, correlations. This is illustrated in the posterior correlations (Figure 3) and covariance matrices (Figure S4 and Table S2). Please note also that we inform the reader about this in the main manuscript in Section 2.4 (p.13 l.2) and that the updated Figure 3 will make this clearer. Although we now mention the importance of posterior correlations in the main manuscript (p.14 l.29 and p.16 l.15),

10  we have preferred to discuss the details of the posterior correlations only in the Supplementary Information (section S7). We think that this discussion is of interest to firn model developers but probably not to the majority of the firn model users. We hope that discussing more our prior distributions at (p.12 l.40), at (p.13 l.19), at (p.14 l.29) in the caption of Table 1 and in the Supplementary Information (section S3) will make our approach more understandable for the reader.

As suggested, we have changed Figure 3 to two-dimensional graphs. We believe that this will help the reader to visualize the multi-dimensionality of the model calibration. Note also that we now explicitly state that the posterior distributions are characterised by some correlation features (p.14 l.29). Figure 3 still provides marginal posterior distributions in parameters but we removed the limits of the 95% credible intervals to avoid overloading the graph

20  (these are still available in Table 1 and can be compared with the weakly informative prior distributions also given in Table 1).

*5.1 Data. Here, I'm with you, I don't want to go through checking each dataset again, but you should at least give some information about these data, citing the spencer paper, and maybe a few other, to illustrate what is the*

25  *uncertainty in the data , given different methods (weighting a full core is not quite the same as doing gamma, or CT..).*

We modify the manuscript in order to refer the reader to our Data Availability section and to our Supplementary Information as soon as we introduce the dataset we use (p.10 l.32). We also add a paragraph dedicated to measurement uncertainty for firn depth-density profiles in Section 2.1 (p.11 l.9). In this paragraph, we refer to several studies that

30  investigate the magnitude of measurement uncertainty.
It is important to highlight that we selected these 91 cores carefully. It was a meticulous effort to look at every profile individually and ruling it as acceptable or not. Unfortunately, even the most recent density measurements do not come with any uncertainty estimation. Thus, the selection relied on our appreciation of the quality of the data. The SUMup dataset has more than 1000 cores and simply taking all these cores was not possible (even though it would have been

35  an easy solution). We also rejected more than half of the Spencer dataset, and we explored other data sources. We know that the selection criteria "accepted by the authors" is not ideal. However, we believe that this was the best solution given the firn core data available.
In our approach, the variances of the likelihood function can mitigate consequences of measurement errors on the calibration process. Also, using DIP as a metric reduces the impact of single measurement errors. Since DIP quantifies

40  a smoothed depth-density profile, individual errors in point measurements of density in depth should (1) have a minor impact on DIP and (2) average out. Also, the use of 69 cores in the calibration should mitigate the effect of a bias in anyone of the 69 cores. Finally, the percentages of DIP used for the calculations of variances are consistent (and even conservative) with respect to the uncertainty estimates of the studies referenced at (p.11 l.10).

45  *5.2 Metric. Is DIP really the best metric? Is it faithful? You are the first to use this metric for the calibration of firn air models. Everyone else was using rho(z) directly. Does it give the same answer? You should demonstrate that. Is it a good metric also for other applications of firn modelign, such as the close-off depth estimation? A paragraph demonstrating the usefullness and validity of this metric would be nice.*

50  We have added a paragraph to discuss our choice of the DIP metric in Section 2.1 (p.10 l.49). In this paragraph, we highlight that DIP is a metric commonly used by firn modellers and by the firn science community in general. Using all individual $\rho(z)$ measurements has several drawbacks: a much stronger sensitivity to individual errors and the fact that the number of $\rho(z)$ values per core is very variable and dependent on the measurement technique. The

approach of Herron and Langway (1980) had been considered. They asserted that the slope of $ln\left(\frac{\rho}{\rho_i-\rho}\right)$ is linear in depth for both stages and thus tuned the modelled slopes to the observed slopes. However, by looking at these slopes for the firn cores, we estimated that this is a very crude approximation. Other authors have calibrated with respect to the depth at which the firn reaches 550 and 830 kg m$^{-3}$ density. This has the drawback that it does not capture the shape of the depth-density profile. Another solution could have been to smooth the $\rho(z)$ profiles (which is the approach of Spencer et al. (2001) and Morris (2018)). By integrating the porosity in depth, DIP is a metric comparable to this approach. We hope that these points and the paragraph added to the manuscript address the concerns of the reviewer with respect to the DIP metric.

*The general objective of the paper is, if I quote the abstract to "demonstrate how modeland parameter-related uncertainties potentially affect ice sheet mass balance assessments". It's a great idea, and such a thorough assessment of firn models has not been done. Lundin et al. 2017 highlighted some model deficiencies, but did nothing to remedy those. I support a future version of this paper to be published with a quantitative answer to this question, but we are not there yet.*

The reviewer pointed out in this remark that some statements in the first version of the paper were inadequate with respect to its general objective. This study is focused on the methodology of a Bayesian calibration process applied to firn models. Furthermore, we show how uncertainties derived from this method have potential impacts for the broader firn science community. In order to clarify this, we remove the statement "how model- and parameter-related uncertainties potentially affect ice sheet mass balance assessments" from the abstract. For the same reasons, we also removed the section about a Greenland-wide estimate of firn compaction uncertainty from the Discussion. We deemed it too speculative given that many other factors must be taken into account for such an estimation (most notably the impact of melting outside the dry snow zone, climatic gradients in topographically complex areas, geographical patterns of surface density and satellite measurements accuracy). While fully evaluating uncertainty in ice sheet mass balance assessments is beyond the scope of our study, we strongly believe that our work is an important step towards this objective. We are currently working in this direction and we hope to submit an ice-sheet wide uncertainty study in the future, however we agree with the reviewer that "we are not there yet".

As mentioned above, the section removed from the Discussion has been replaced by another section (p.16 l.49). The latter aims at demonstrating the usefulness of Bayesian posterior uncertainty evaluations in terms of (1) compaction rates and (2) age of firn at pore close-off.

*If I dont want to run the community firn model (CFM) 30,000 times but just a few, to get a gist of the uncertainty in my specific application, should I use a wide range of (T, accum) scenarios (step 1)? or rather use different model physics (step 3)? Or one of the models, but with a range of parameters (step 4)? This is a practical question that would be really useful to future users of the community firn model.*

The modifications to the manuscript offer two different ways to address the practical problem of running 30 000 simulations for assessments of parameter-related and model-related uncertainty (steps 3 and 4). The first, and more straightforward, approach is to use the coefficients of variation we give (p.17 l.10, p.17 l.13 and Table 3) when discussing uncertainty in computed compaction rates and ages of firn at pore-close off. These figures provide an approximation of the typical uncertainty one can expect due to uncertainty in parameter values and in choice of model. We believe that the coefficients of variations, expressed in%, provide a straightforward overview to the reader of the uncertainty one can expect in firn model output. The second approach requires slightly more work and more computational effort, but it is more rigorous. We introduce the notion of the normal approximation to the posterior densities in Section 2.4 (p.14 l.9). The normal approximation to the posterior is a practical and commonly used solution in such circumstances (see Gelman et al. (2013)). Any firn-model user can sample a high number of parameter combinations from these normal distributions. Firn model output based on this sample can subsequently be used to build uncertainty intervals for any application. All the details and information required for generating samples of parameter combinations are provided in the Supplementary Information (section S6). The question of how large the generated sample should be cannot be answered unequivocally. Statistical theory tells that the uncertainty intervals (e.g. at 95% precision level or any other) will converge to the true intervals as the sample size increases. We used 500 samples in the study but it was clear that at most sites, the uncertainty intervals were very close to those reached with much fewer samples. One can start

computing uncertainty intervals with a certain number of samples (e.g. 50) and evaluate how intervals change with larger number of samples. A good rule of thumb is that an optimal sample sized is reached once uncertainty intervals remain stable with a further growth in the sample size.

Concerning the climate-related uncertainty (step 1), we believe that this is beyond the scope of this study. We have integrated climatic noise, which allows propagation of climatic uncertainty in the estimation of uncertainty intervals of parameter values. As such, the range of parameter values used in computing the coefficients of variations and characterising the posterior distributions incorporates parameter values influenced by climatic noise. Estimating the plausible range of climatic forcing at any given location in Greenland or Antarctica must be addressed by studies dedicated to regional climate models. Interestingly, many intercomparisons of climate models have been submitted and/or published recently.

*And finally, when you have done that, it would be great to go back and recalculate the uncertainty in mass balance from altimetry, using the above mentionned decomposition. This is the great paper I'd like to read. It's marginally more work from what you have done, running a few more simulations on the same framework, but I think it would be really worth it.*

We discuss this aspect in our response above. We believe that this study is a step towards this direction and our long-term objective is indeed to "*recalculate the uncertainty in mass balance from altimetry*". However, a thorough calculation requires taking many other aspects into account and goes much further than the methodology we present in this study. We work thoroughly on this long-term objective and we welcome the interest of the reviewer in this subject.

*Method: Why did you go for Monte-Carlo, rather than a form of generalized least squares, which would have converged in <10 runs very likely, and could have easily dealt with covariance in model parameters? I agree that this problem is non linear, and underdetermined, but it is also monotonic, so least squares are applicable, and converge much faster. That being said, your method is valid.*

The reviewer is correct that multivariate least squares regression would have required less model runs and would probably have reached close "best fit" parameter estimates. We have decided to follow the Bayesian MCMC approach for two reasons. First and foremost, it is much more powerful to assess uncertainty in parameter values given the available data. This point is of major importance to our study. Secondly, it provides a robust framework for future developments. In the future, it will be desirable to integrate other type of data to firn model calibration (e.g. strain rates and radar reflection if such data become publicly available) and to apply such a calibration method to models of greater complexity which exhibit strongly non-linear behaviour (e.g. ice sheet models, climate models or even firn models that become more complex). Bayesian MCMC are a powerful tool in such circumstances. Finally, there is scope to reduce computational costs as recent algorithmic developments (such as Hamiltonian Monte Carlo) allow faster posterior convergence.

*choice of models. Why did you choose the Arthern model as one of your 2 models, rather than the IMAU versions (ligtenberg or kuipers)? We already know that the physical formulation of Arthern is not right (Lundin 2017). You later discuss the IMAU version. I think it would make more sense to publish an optimisation for these rather than the Arthern, which shows no sensitivity to the accumulation rate, something we know is wrong.*

We discussed the choice of models in length before carrying out this study. Finally, we favoured to focus on the Arthern model because it is the original formulation of the Ligtenberg and Kuipers Munneke versions, and also in order to demonstrate that its inherent flaws can be corrected through our calibration method. Model sensitivities can be evaluated by derivative calculations. We have added a paragraph in the Discussion section (p.16 l.19) where we show that the over-sensitivity of the Arthern model to accumulation rates is appropriately rectified.

[revised manuscript text omitted]

In order to let uncertainty in RACMO2 output affect the calibration process, we perturb the temperature and accumulation time series that serve as climatic forcing for the firn models. At each iteration (a round of simulations with a given parameter set at all the calibration sites) and for each individual calibration site, we randomly draw an individual climatic perturbation value $c_p$ from a standard Normal distribution (Eq. (S1)). As such, every calibration site has a specific $c_p$ value, which changes at each iteration. We use observed statistics of RACMO2 errors in temperature and Surface Mass Balance to determine the perturbation.

For GrIS, Noël et al. (2019) report RMSE values with respect to field observations for temperature and surface mass balance flux of 2.1 K and 69 m w.e. yr$^{-1}$ respectively (in their Supplementary Material).

Each monthly value of the RACMO2 time series is therefore perturbed by the corresponding RMSE value scaled by $c_p$ (Eq. (S2), (S3), (S4)).

We favour this approach rather than drawing a different random perturbation at each time step. The latter method would cause perturbations of opposite signs to occur on a very short timescale, which would result in unrealistic short term climatic variability (e.g. a very warm perturbation could be immediately followed by a very cold perturbation in the next month). Also, using the same $c_p$ value to quantify the magnitude of the perturbation for temperature and accumulation preserves the strong correlation between these variables. Warm (cold) temperature perturbations coincide with high (low) accumulation perturbations, which keeps our random perturbations physically plausible.

The part of the total accumulation perturbation attributed to each monthly time step is weighted by the actual accumulation at that time step. This attributes larger absolute noise in accumulation to high-accumulation months and lower absolute noise to low-accumulation months (Eq. (S3), (S4)).

Our approach is summarized in Eq. (S1), (S2), (S3) and (S4). These equations are applied at all iterations of the calibration process.

$$c_p \sim N(0,1) \quad \text{at all calibration sites} \tag{S1}$$

$$T_t^* = T_t + c_p \sigma_T \quad \text{at all } t \tag{S2}$$

$$\dot{b}_{tot}^* = n_{yr} c_p \sigma_{SMB} \tag{S3}$$

$$\dot{b}_t^* = \dot{b}_t + \dot{b}_{tot}^* \frac{\dot{b}_t}{\sum_t \dot{b}_t} \tag{S4}$$

where $T_t$ and $\dot{b}_t$ are temperature and accumulation rate as computed by RACMO2 at time step $t$ and the $*$ superscript denotes the perturbed quantity. $n_{yr}$ is the total number of years in a given simulation, $\dot{b}_{tot}^*$ is the total accumulation perturbation applied for that simulation and $\sigma_T$ and $\sigma_{SMB}$ are the temperature and surface mass balance flux RMSE values (as mentioned above, $\sigma_T = 2.1$ K and $\sigma_{SMB} = 69$ m w.e. yr$^{-1}$ for GrIS). Note that by using a RMSE value on the surface mass balance flux, we overestimate uncertainty because the observed RMSE is mostly driven by errors in melt amounts which do not apply at the sites of our dataset, all from the dry snow zone area. For AIS, we apply the exact

same process for perturbing the temperature variables. We use the RMSE value reported by van Wessem et al. (2018) and set $\sigma_T = 1.3$ K. The accumulation conditions of AIS forces the use of a slightly different method for perturbing the accumulation rate. In terms of magnitude, RACMO2 errors are much larger in coastal areas, where accumulation rates are high. In contrast, in the dry interior of the ice sheet where most of the cores of our dataset come from, the magnitude of RACMO2 errors is small due to low accumulation rates. As such, applying noise based on the ice sheet wide RMSE value would result in noise signals larger than actual accumulation values at most of our dry sites. We thus use the average RACMO surface mass balance bias of 5% (van Wessem et al., 2018) as a proxy for one standard deviation. For AIS, Eq. (S3) and (S4) are replaced by Eq. (S5).

$$\dot{b}_t^* = \dot{b}_t + 0.05\, c_n\, \dot{b}_t \tag{S5}$$

As explained in Sect. 2.2, we also let uncertainty in fresh snow density, $\rho_0$, affect the calibration process by applying random perturbations to each $\rho_0^t$. In contrast to the climatic perturbation, the perturbation in $\rho_0$ must not be iteration specific but can be specific to each single time step $t$. Indeed, it is not unrealistic that a month with anomalously low fresh snow density is immediately followed by a month of anomalously high fresh snow density for example. We determine surface density values at each site from the firn cores of our dataset, $\rho_0^{core}$, and we perturb these values based on a standard deviation of 25 kg m$^{-3}$. As such, adding noise to $\rho_0$ simplifies to Eq. (S6).

$$\rho_{0,t}^* \sim N(\rho_0^{core}, 25) \tag{S6}$$

We emphasize that the aim of this study is not to conduct a complete sensitivity analysis of firn densification to climatic forcing and to fresh snow density. The objective of the perturbations is to let reasonable estimates of errors in those fields to be accounted for in the calibration process.

**S3 Prior correlations in HL and Ar**

The Arrhenius form of HL and Ar (Eq. (4) and (5)) allows us to include some correlation in the prior distributions over the parameters of these models. The values of the Arrhenius pre-exponential factors ($k_0^*, k_1^*, k_0^{Ar}$ and $k_1^{Ar}$) are correlated with their corresponding activation energies ($E_0, E_1$ and $E_g$). For any given constant temperature, modelled densification rates, $\frac{d\rho}{dt}$, can be kept constant despite a change in the pre-exponential factor if the corresponding activation energy is changed accordingly and vice versa. As such, changes in these parameters can potentially compensate in the calculation of $DIP$ values and of the likelihood function (Eq. (8)).

By enforcing constant $\frac{d\rho}{dt}$, exact compensation is ensured by the following equalities:

$$k_{0,mv}^* = k_{0,HL}^* \exp\left(\frac{E_{0,mv} - E_{0,HL}}{R\,T}\right) \tag{S7}$$

$$k_{1,mv}^* = k_{1,HL}^* \exp\left(\frac{E_{1,mv} - E_{1,HL}}{R\,T}\right) \tag{S8}$$

$$k_{0,mv}^{Ar} = k_{0,Ar}^{Ar} \exp\left(\frac{E_{g,Ar} - E_{g,mv}}{R\,T}\right) \tag{S9}$$

$$k_{1,mv}^{Ar} = k_{1,Ar}^{Ar} \exp\left(\frac{E_{g,Ar} - E_{g,mv}}{R\,T}\right) \tag{S10}$$

where $HL$ and $Ar$ subscripts denote the original values in HL and Ar, and the $mv$ subscript denotes a modified value of the parameter. Firstly, we generate 10000 random values of temperature $T$ in the range of annual mean temperatures covered by our dataset. Secondly, for each random temperature, we generate random values of $E_{0,mv}$, $E_{1,mv}$ and $E_{g,mv}$ in an interval of $\pm 500$ J mol$^{-1}$ around the original values. Thirdly, we calculate the corresponding values in the pre-exponential factors from Eq. (S7), (S8), (S9) and (S10). This results in 10000 pairs of $(k_{0,mv}^*, E_{0,mv})$, $(k_{1,mv}^*, E_{1,mv})$, $(k_{0,mv}^{Ar}, E_{g,mv})$ and $(k_{1,mv}^{Ar}, E_{g,mv})$, from which we calculate correlation coefficients. The absolute values of all four correlation coefficients lie in the interval [0.75; 0.78]. We decide to fix all prior correlation coefficients to -0.75 (HL parameters, negatively correlated) and 0.75 (Ar parameters, positively correlated). The process described necessarily results in perfectly correlated $k_{0,mv}^{Ar}$ and $k_{1,mv}^{Ar}$. We also set the prior correlation between these parameters to 0.75.

We emphasize here that any other pair of *a priori* uncorrelated parameters can certainly be correlated *a posteriori* if the calibration process identifies such quantitative behaviour when the observed data is considered. This is highlighted and further discussed in Sect. S7.

**S4 The likelihood function, Eq. (8)**

[revised manuscript text omitted]

**S6 Normal approximation to the posterior**

The ensembles of parameter combinations obtained for each model provide large samples, representative of the posterior probability distributions over their respective parameter space. The most efficient way to assess parameter-related uncertainty is to run a model with a high number of random parameter combinations from these ensembles, which is demonstrated in Sect. 3. However, this means that for any firn modelling study, access must be easy to such posterior ensembles or an MCMC algorithm must be re-executed. To circumvent these practical difficulties, it is approximately correct to sample random parameter combinations from a multivariate normal distribution centred about the mean of the posterior ensemble and with covariance matrix set to the posterior ensemble covariance matrix. This is commonly referred to as a normal approximation to the posterior (Gelman et al., 2013). Table S2 provides both the posterior mean and posterior covariance for the HL, Ar and LZ models.

We assess how random samples from the normal approximations compare to samples from the posterior ensembles in Fig. S4. Posterior samples and the normal approximations are very similar, with correlations only slightly less well captured in the tails of the distributions. It results in a slight overestimation of uncertainty and thus conservative estimates of uncertainty. This has been confirmed by additional model simulations with values sampled from the normal approximations (not shown).

**S4 Compaction anomaly calculation**

[revised manuscript text omitted]

**Table S1.** The 91 firn core dataset used in this study. * symbols indicate the core is part of the evaluation data. Lat and Lon designate latitude and longitude respectively. Year indicates the year of drilling of the core. $\dot{b}$ is the accumulation rate. T is the temperature. $\rho_0$ is the surface density boundary condition that was derived individually for each core by extrapolating density measurements until the surface (random noise is added to $\rho_0$ as discussed in Sect. S2). Var designates the site-specific variance used for the terms of $\Sigma_{15}$ and $\Sigma_{pc}$ (see Text S2 S4 for their calculation). The core spencer90 has only a single density measurement above 15 m depth and its $DIP15$ is discarded.

| Parameters | Posterior mean | Posterior covariance matrix |
|---|---|---|
| HL | $k_0^*, k_1^*, E_0,$ $E_1, a, b$ | $\begin{bmatrix} 16.7, 649, 10760, \\ 21000, 0.88, 0.66 \end{bmatrix}$ | $\begin{bmatrix} 34.4 & 40.2 & 4500 & 324 & -0.0685 & -0.0195 \\ 40.2 & 44000 & 618 & 161000 & 1.087 & -3.670 \\ 4502 & 618 & 710000 & 7080 & -29.95 & 1.94 \\ 324 & 1610000 & 7080 & 694000 & 7.86 & -27.51 \\ -0.0685 & 1.087 & -29.95 & 7.86 & 0.0051 & -0.0012 \\ -0.0195 & -3.670 & 1.94 & -27.51 & -0.0012 & 0.0036 \end{bmatrix}$ |

| Parameters | Posterior mean | Posterior covariance matrix |
|---|---|---|
| HL — $k_0^*, k_1^*, E_0, E_1, a, b$ | $[16.7, 649, 10760, 21000, 0.88, 0.66]$ | $\begin{bmatrix} 34.4 & 40.2 & 4500 & 324 & -0.0685 & -0.0195 \\ 40.2 & 44000 & 618 & 161000 & 1.087 & -3.670 \\ 4502 & 618 & 710000 & 7080 & -29.95 & 1.94 \\ 324 & 1610000 & 7080 & 694000 & 7.86 & -27.51 \\ -0.0685 & 1.087 & -29.95 & 7.86 & 0.0051 & -0.0012 \\ -0.0195 & -3.670 & 1.94 & -27.51 & -0.0012 & 0.0036 \end{bmatrix}$ |
| Ar — $k_0^{Ar}, k_1^{Ar}, E_g, \alpha, \beta$ | $[0.080, 0.028, 40900, 0.78, 0.69]$ | $\begin{bmatrix} 5.62\ 10^{-4} & 1.55\ 10^{-4} & -12.66 & 9.65\ 10^{-5} & -3.23\ 10^{-4} \\ 1.55\ 10^{-4} & 7.41\ 10^{-5} & -4.64 & -2.04\ 10^{-4} & 1.05\ 10^{-4} \\ -12.66 & -4.64 & 360000 & 11.0 & 4.67 \\ 9.65\ 10^{-5} & -2.04\ 10^{-4} & 11.0 & 3.30\ 10^{-3} & -1.01\ 10^{-3} \\ -3.23\ 10^{-4} & 1.05\ 10^{-4} & 4.67 & -1.01\ 10^{-3} & 3.12\ 10^{-3} \end{bmatrix}$ |
| LZ — $lz_a, lz_b, lz_{11}, lz_{12}, lz_{13}, lz_{21}, lz_{22}, lz_{23}$ | $[7.56, -2.091, -14.71, 7.269, -1.019, -1.513, 6.0203, -0.09127]$ | $\begin{bmatrix} 5.27 & -0.198 & -1.20 & -1.68 & -0.0239 & 5.53\ 10^{-3} & -0.0606 & 4.13\ 10^{-3} \\ -0.198 & 0.0116 & 0.218 & -0.0612 & 0.0134 & -0.0158 & -2.29\ 10^{-3} & -7.37\ 10^{-4} \\ -1.20 & 0.218 & 14.6 & -3.96 & 0.801 & 0.368 & 0.354 & 0.0129 \\ -1.68 & -0.0612 & -3.96 & 13.3 & -0.309 & -0.0850 & 5.40 & 0.0166 \\ -0.0239 & 0.0134 & 0.801 & -0.309 & -0.0502 & -0.0173 & 0.0252 & -4.42\ 10^{-4} \\ 5.53\ 10^{-3} & -0.0158 & 0.368 & -0.0850 & -0.0173 & 0.446 & -0.429 & 0.0131 \\ -0.0606 & -2.29\ 10^{-3} & 0.354 & 5.40 & 0.0252 & -0.429 & 3.94 & -2.59\ 10^{-4} \\ 4.13\ 10^{-3} & -7.37\ 10^{-4} & 0.0129 & 0.0166 & -4.42\ 10^{-4} & 0.0131 & -2.59\ 10^{-4} & 4.80\ 10^{-4} \end{bmatrix}$ |

**Table S2.** The posterior means and covariance matrices for the free parameters of HL, Ar and LZ. These statistics can be used to generate random parameter combinations following a normal approximation.

| $cmp_{an}^{00-17}$ [m] | HL | $MAP_{HL}$ | Ar | $MAP_{Ar}$ | LZ | $MAP_{LZ}$ |
|---|---|---|---|---|---|---|
| EGRIP | 0.626 | 0.648 | 0.735 | 0.656 | 0.675 | 0.695 |
| Summit | 0.464 | 0.461 | 0.442 | 0.442 | 0.481 | 0.475 |
| id359 | 0.748 | 0.763 | 0.866 | 0.781 | 0.815 | 0.837 |
| id369 | 0.609 | 0.587 | 0.597 | 0.574 | 0.623 | 0.617 |
| id373 | 0.613 | 0.637 | 0.735 | 0.649 | 0.664 | 0.693 |
| id385 | 0.319 | 0.333 | 0.407 | 0.356 | 0.373 | 0.393 |
| id423 | 0.528 | 0.559 | 0.649 | 0.566 | 0.565 | 0.600 |
| id514 | 0.358 | 0.367 | 0.409 | 0.378 | 0.409 | 0.423 |
| id531 | 0.336 | 0.333 | 0.304 | 0.328 | 0.373 | 0.371 |
| id534 | 0.216 | 0.214 | 0.189 | 0.205 | 0.229 | 0.237 |
| Basin8 | 0.764 | 0.760 | 0.728 | 0.671 | 0.641 | 0.611 |
| D2 | 0.244 | 0.255 | 0.222 | 0.164 | 0.158 | 0.133 |
| D4 | 0.560 | 0.553 | 0.550 | 0.465 | 0.500 | 0.479 |
| HumboldtM | 0.630 | 0.616 | 0.606 | 0.534 | 0.562 | 0.545 |
| NASAE1 | 0.591 | 0.568 | 0.578 | 0.555 | 0.605 | 0.596 |
| spencer6 | 0.380 | 0.386 | 0.391 | 0.367 | 0.393 | 0.399 |
| spencer16 | 0.499 | 0.489 | 0.500 | 0.428 | 0.462 | 0.448 |
| spencer17 | 0.503 | 0.534 | 0.611 | 0.538 | 0.536 | 0.566 |
| spencer66 | 0.963 | 0.944 | 0.955 | 0.912 | 0.926 | 0.889 |
| spencer67 | 0.832 | 0.812 | 0.821 | 0.762 | 0.761 | 0.744 |
| spencer68 | 0.895 | 0.875 | 0.895 | 0.831 | 0.838 | 0.811 |
| spencer69 | 0.935 | 0.915 | 0.920 | 0.880 | 0.895 | 0.857 |
| spencer70 | 1.078 | 1.056 | 1.060 | 1.017 | 1.025 | 0.982 |
| spencer71 | 1.094 | 1.066 | 1.126 | 1.042 | 1.107 | 1.093 |
| spencer72 | 0.809 | 0.806 | 0.852 | 0.780 | 0.812 | 0.799 |
| spencer73 | 0.793 | 0.786 | 0.835 | 0.756 | 0.782 | 0.763 |
| spencer74 | 0.856 | 0.842 | 0.861 | 0.785 | 0.786 | 0.771 |

**Table S2.** Total compaction anomaly over 2000-2017 ($cmp_{an}^{00-17}$) at the 27 GrIS sites. See text S3 for calculation details.

**Figures**

[Figure]

**Figure S1.** Climatic conditions at the 91 sites of the dataset

[Figure]

**Figure S2.** Quantiles-Quantiles plots for the errors of the three original models (HL, Ar, LZ) computed on the entire dataset. The alignment of the points along the red line informs about the fit to a normal distribution.

[Figure]

**Figure S3.** Sampling chains of each parameter for (a) HL, (b) Ar, (c) LZ. The x-axis displays the iteration number, the y-axis displays the parameter value. The dashed pink line shows the value of the original model, which is also the starting point of each chain.

[Figure]

**Figure S4.** Evaluation of the normal approximations to the posterior distributions for (a) HL, (b) Ar, (c) LZ. Where possible, correlated parameters share a same graph.

**(a) HL model**

| | $k_0^*$ | $k_1^*$ | $E_0$ | $E_1$ | $a$ | $b$ |
|---|---|---|---|---|---|---|
| $k_0^*$ | 1 | 0.03 | 0.91 | 0.07 | -0.16 | -0.06 |
| $k_1^*$ | | 1 | 0 | 0.92 | 0.07 | -0.29 |
| $E_0$ | | | 1 | 0.01 | -0.5 | 0.04 |
| $E_1$ | | | | 1 | 0.13 | -0.55 |
| $a$ | | | | | 1 | -0.28 |
| $b$ | | | | | | 1 |

**(b) Ar model**

| | $k_0^{Ar}$ | $k_1^{Ar}$ | $E_g$ | $\alpha$ | $\beta$ |
|---|---|---|---|---|---|
| $k_0^{Ar}$ | 1 | 0.76 | -0.89 | 0.07 | -0.24 |
| $k_1^{Ar}$ | | 1 | -0.9 | -0.41 | 0.22 |
| $E_g$ | | | 1 | 0.32 | 0.14 |
| $\alpha$ | | | | 1 | -0.32 |
| $\beta$ | | | | | 1 |

**(c) LZ model**

| | $lz_a$ | $lz_b$ | $lz_{11}$ | $lz_{12}$ | $lz_{13}$ | $lz_{21}$ | $lz_{22}$ | $lz_{23}$ |
|---|---|---|---|---|---|---|---|---|
| $lz_a$ | 1 | -0.8 | -0.14 | -0.2 | -0.05 | 0 | -0.01 | 0.08 |
| $lz_b$ | | 1 | 0.53 | -0.16 | 0.56 | -0.22 | -0.01 | -0.31 |
| $lz_{11}$ | | | 1 | -0.28 | 0.94 | 0.14 | 0.05 | 0.15 |
| $lz_{12}$ | | | | 1 | -0.38 | -0.03 | 0.74 | 0.21 |
| $lz_{13}$ | | | | | 1 | -0.12 | 0.06 | -0.09 |
| $lz_{21}$ | | | | | | 1 | -0.32 | 0.9 |
| $lz_{22}$ | | | | | | | 1 | -0.01 |
| $lz_{23}$ | | | | | | | | 1 |

**Figure S5.** Posterior correlation matrices.

---

## Editor Decision (ED1)

I would like to thank both reviewers for their constructive comments on this manuscript and also the authors for posting their response to the reviewers' comments.

The original manuscript provides a detailed analysis of the factors affecting firn densification modelling, but the reviewers raise a number of issues that should be addressed to confirm the robustness of the results.

In their response, the authors outline the steps that they propose to take to address the points raised by the reviewers. These include providing clearer justification for their choice of performance metric and the completion of additional simulations that will explore the sensitivity of the results to boundary conditions, including climate forcing, and model assumptions. I also encourage the authors to consider steps they could take to make their findings more useful to the wider glaciological community, for example by providing a quantitative assessment of different sources of uncertainty – as suggested by one of the reviewers.

In general, both reviews are positive, and I therefore encourage the authors to submit a revised manuscript that addresses the points raised during the preliminary review process.

Kind regards,

Pippa Whitehouse

---

## Author Response (AR2)

Dear Pippa Whitehouse,

We thank you very much for your comments and insights on the manuscript. We have addressed all your comments one by one and made the necessary modifications in the updated version of the manuscript. Please find below our responses (in blue) to the comments (in black). A marked-up version of the updated manuscript is provided for your convenience below the responses. Throughout our responses, page and line numbers relate to this document. Should you require any further information, please do not hesitate to contact me.

On behalf of all authors,

Vincent Verjans

Minor points – page/line numbers relate to version2 of the manuscript (no track changes)

p.1 l.31: most points in this paragraph document why each application is important rather than recommending specific steps for improvement – the text on line 31 is a little misleading since it could be taken to suggest that you will investigate sensitivity to climate conditions in this study

We have slightly changed the phrasings in the paragraph to underline the direct role of firn models in these applications. We have also removed the mention to sensitivity to climatic conditions in order to avoid any misunderstanding. However, it should be noted that firn models simulate densification as a function of climate, and their inherent objective is thus to capture the climatic sensitivity of the densification process.

Changes in the text (p.8):

"Consequently, uncertainties in modelled densification rates have a direct impact on mass balance estimates, which rely on a correct conversion from measured volume changes to mass changes"

"Model estimates of current and future surface mass balance of the AIS and GrIS are thus dependent on accurate models of firn evolution."

p.2 l.16: check throughout whether it is appropriate use to use 'AIS/GrIS' or 'the AIS/GrIS'

Indeed, the previous manuscript was inconsistently alternating between both wordings. We now exclusively use "the AIS/GrIS" when used as a noun (e.g. "locations of the AIS"). We keep the use of "AIS/GrIS" when used as an adjective (e.g. "14 AIS cores").

p.4 l.5-7: text seems a little out of place here, information would fit better at the end of the final paragraph of p.3

We moved the paragraph as suggested (p.11 l.1).

p.4 l.25-26: the statement that you use a constant site-specific value is slightly at odds with the statement on lines 29-30 that you add random noise at every model time step. It would be useful to mention the approach used to account for uncertainty earlier in this paragraph

We rephrased the paragraph in order to remove the mention to constant values. We also mention our treatment of uncertainty in $\rho_0$ earlier, as suggested (p.11 l.28):

"At each site, the $\rho_0$ value is taken in agreement with the shallow densities measured in the corresponding core of the dataset. However, measurements of fresh snow density are highly variable (e.g. Fausto et al., 2018). We account for uncertainty in this parameter by adding normally distributed random noise with standard deviation 25 kg m$^{-3}$ to $\rho_0$ at every model time step (see Supplementary Information)."

p.6 l.9-11: do Li and Zwally (2015) use a different formulation of the equation, or do they just determine different parameter values compared with Li and Zwally (2011)?

The Li and Zwally (2015) model uses a formulation very close, but still different to the Li and Zwally (2011) model. We clarified this aspect in the manuscript (p.13 l.14):

" Later, Li and Zwally (2015) developed a densification model calibrated for Antarctic firn. The latter model uses the same governing equations as LZ for $c_0$ and $c_1$ but different formulations for $\beta_0$ and $\beta_1$ (Eq. (6))."

p.6 l.25: to data -> with data

Changed (p.14 l.2)

p.7 l.23: several steps are described prior to the mention of figure 2a; please indicate how the text on lines 20-23 relates to steps shown in figure 2. Also, please clarify whether calculations are carried out for both θi and θi* on each iteration

The confusion stems from the first iteration (at $i = 0$), which must be executed to start the Random Walk Metropolis algorithm. We have clarified that the first step must compute an initial posterior probability distribution for the algorithm to start because the acceptance step (Fig. 2e) requires a ratio $P\left(\theta_i^*|Y\right)/P\left(\theta_i|Y\right)$. This first step is executed by using the original models parameter values (from HL, Ar and LZ). This has been included in the description of the algorithm.

We have also emphasised that calculations for $\theta_i$ must not be performed at each iteration, since they were performed at a previous stage. As such, the posterior probability value $P(\theta_i|Y)$ is kept in memory and only $P(\theta_i^*|Y)$ must be computed.

We hope that the adjustments made in the text now provide better clarity (p.14 l.25). We also hope that Figure 2 will be next to the relevant text after the typesetting.

p.8 l.1: variance -> covariance (as defined on l.25 of the previous page)

Changed (p.15 l.10)

p.8 l.16: 'a 500 random sample' – rephrase

Changed (p.15 l.25): "from 500 random samples"

p.9 l.22: the DML plots are figs. 5g-i

Thank you for pointing this out. The text has been changed (p.16 l.30).

p.9 l.28: 'the better performance at the GrIS evaluation sites…' – make it clear that this text relates to the performance of the original model

Changed (p17. l.5): "As such, the better performance at the GrIS evaluation sites of the original HL is likely due to its parameterisation being better suited for the particular temperature range corresponding to the conditions of the latter sites."

p.10 l.7: it is not clear to me how the LZ dual model was constructed; do you determine different parameters for each ice sheet by dividing the calibration data set, or is the whole formulation of the model different? Refer to Table 2 when quoting results for the LZ dual model

The whole formulation of the model is different. (The formulations are close but not the same). We hope that our changes related to the comment above clarifies this aspect.

Furthermore, we clarified the construction of LZ dual (p.17 l.16): " We compute results at the AIS and GrIS evaluation sites using the Li and Zwally (2015) model for the AIS and the Li and Zwally (2011) model for the GrIS, so that both models are applied to the ice sheet for which they were originally developed. We call this pairing of models LZ dual and evaluate its general performance."

We now refer to Table 2 when stating the RMSE values of LZ dual (p.17 l.21).

p.10: if feasible, it would be useful to include a figure showing the results for LZ dual and IMAU-FDM (e.g. similar to figure 4) in the supplementary information

We have added similar scatter plots for the LZ dual and IMAU models in the Supplementary Information: Figure S6. We inform the reader about this in the main text; in the caption of Figure 4, we added:

"Similar scatter plots for the LZ dual and IMAU results are shown in the Supplementary Information (Fig. S6)."

p.10 l.23: please include information on how uncertainty intervals were constructed in the captions to figures 4 and 5

We added in both captions: "The 95% credible intervals are computed from results of 500 randomly selected parameter combinations from the posterior ensembles of each model (HL, Ar, LZ)."

p.11 l.5: 'indicate a weaker increase…' – weaker than what?

We specified (p.18 l.16): "Our results of stage-1 exponents $(a, \alpha)$ smaller than 1 indicate a weaker increase in densification rates with pressure than assumed in the original versions of Ar and HL."

p.11 l.12: 'The same can be applied…' – not clear what 'The same' refers to

Changed (p.18 l.23): "The difference in sensitivities of stage-1 and stage-2 densification to accumulation also holds in the LZ model (…)"

p.11, l.13: please refer to a figure or table when quoting correlation coefficient values

We now refer to the Figure S5 of the posterior correlation matrices (p.18 l.26).

p.11 l.22: over-sensitivity -> over-sensitivity in Ar

Added (p.18 l.3)

p.12 l.25: what is the reference period? If it is 2000-2017 this should be explicitly stated

The reference periods for both ice sheets are different and the information is provided in Section 2.2. Because this section is in the earlier parts of the manuscript, we agree that it is better to remind the reader about it. We have added a reference to Sect. 2.2 in the text (p.20 l.6): "over the reference period (see Sect. 2.2)."

p.12 l.33: make it clearer that uncertainties in the following sentences are calculated using the CV values quote above/ in table 3

We clarified this point by mentioning our use of CV to estimate uncertainty ranges before providing the uncertainty values (p.20 l.14):

" By using the CV values, we can calculate reasonable uncertainty estimates for $cmp_{an}$ and $age_{pc}$. For instance, in the dry snow zone of GrIS, simulated compaction anomalies are typically around 20 cm over 2000-2017, and thus come with an uncertainty of the order of ±4 cm. (…)"

p.13 l.5: a couple of clarifications needed: (i) what does 'it' refer to, and (ii) what does 'Such numbers' refer to?
We changed "it" to specify that we refer to the spatial aggregation of uncertainties (p.20 l.20):
"Absolute uncertainty is thus reduced but still critical given the large area of the AIS over which uncertainties are aggregated when mass balance trends are evaluated."
We changed our use of "Such numbers" in accordance with the previous comment (p.20 l.20):
" The uncertainty ranges calculated from the CV values provide an order of magnitude of errors in firn model outputs that must be accounted for in altimetry-based mass balance assessments and in ice core studies, respectively."

p.13 l.7: the purpose of the text (paragraph?) starting on this line is initially unclear. For example, it is not clear what you mean by 'the different sensitivities...'. You mention that compaction is sensitive to variability and 'general increases' in temperature and accumulation – can you be more explicit about the climate at the two sites, perhaps by including site-specific RACMO2 output in figure 7?
We changed the starting sentence of the paragraph to relate it more closely to the topic of the study. We explicitly mention that we look at the effect of different models and different parameterisations on firn model output and we removed the terms "different sensitivities" which were too vague (p.20 l.23):
"We further investigate how using different models and different parameterisations leads to discrepancies in the modelled compaction. We compute monthly values of compaction anomalies over the 2000-2017 period with the original and MAP models of HL, Ar and LZ (Fig. 7)."
As suggested, we included the climatic anomalies at the two sites displayed in Figure 7. These mean anomalies show both a warming and an increase in accumulation at the sites. We explain the computation of the mean climatic anomalies in the caption:
" Mean climatic anomalies are calculated as a difference between mean climatic values over the period 2000-2017 with respect to the reference period 1960-1979, and based on RACMO2 values."
We considered adding the entire time series of RACMO2 anomalies in accumulation and temperature but after due reflection, we preferred not to do so to avoid overloading the figure. In total, the figure would have included the compaction anomaly time series, two climatic anomaly time series and three insets for each site. We believe that adding the mean climatic anomalies to the figure demonstrates that the original HL and the MAP$_{HL}$ models are less sensitive to the general change in climatic conditions. Furthermore, these two models exhibit also much less seasonal variability, which shows that they are also less sensitive to changes in climatic conditions specific to each month (e.g. less sensitive to summer months getting warmer).

p.13 l.14: short-scale -> short-timescale
Changed (p.20 l.31)

p.13 l.26: 'at most sites…uncertainty intervals do not cover observed DIP values' – this is an important result but I did not see it stated/quantified anywhere in the main text

We agree that this important result needs to be mentioned in the text and not only in the conclusion. We have added a couple of sentences at the end of the third paragraph of Section 3 (p.16 l.31):
" However, at a majority of the evaluation sites, the 95% credible intervals computed for the three models do not include the observed value (Fig. 4). This highlights that the governing equations of the models, which intend to capture densification physics, require improvement, and that parameter calibration in itself cannot overcome this shortcoming. "

p.14 l.12-13: this link takes you to a folder which contains several files that are unrelated to this study, are you able to list a source that just links to the firn data?

For the purpose of clarity, we have removed this link and replaced it by the statement (p.21 l.26):
"41 of the 91 firn cores are from the dataset compiled by Matt Spencer 10 (Spencer et al., 2001), which is available upon request."
It is important to note that this dataset is only a compilation of firn cores and the rights are not owned by Matt Spencer himself. The dataset is often used in the literature, including in The Cryosphere, but no consistent link is provided for it.
For examples:
https://doi.org/10.5194/tc-13-845-2019
https://doi.org/10.1029/2017JF004597

Tables 1 and S2: some terminology issues for large/small values, e.g. 9 104 should be 9x104
We have added the multiplicative symbol everywhere necessary in Table 1 and Table S2.

Table 2: explicitly mention RMSE in the caption rather than just 'The errors'
Changed: " The Root Mean Squared Errors (RMSE) are calculated with respect to the observations of depth integrated porosity until 15 m depth and until pore close-off."

Figure 1: what is the difference between a circle and a cross?
Sorry, we forgot to update the full legend when updating the figure after the first review. The crosses are the evaluation sites and the circles are the calibration sites. Figure 1 has been updated.

Figure 2: in the box titled 'If i is multiple of 100' the second $\Sigma$ should be $\Sigma cov$
Yes, Figure 2 has been updated.

Figure 3: please document what the 'posterior samples' are. Do they represent parameters associated with the 500 parameter sets randomly selected from the ensemble of accepted $\theta$?
We added in the caption: " The posterior samples are 500 randomly selected parameter combinations from the posterior ensembles of each model (HL, Ar, LZ)."

Figure 6: mention the difference between the left and right columns in the caption. Also, is it possible to represent AIS and GrIS data points differently, to support statements in main text?
The difference between graphs in the left and right column is now explicitly mentioned in the caption:

"Graphs in the left column display the mean annual temperature on the x-axis and those in the right column display the mean annual accumulation rate."
We have added black contours to the GrIS sites in order to distinguish between AIS and GrIS sites. We hope that this provides better support to the statements in the main text. For example, it shows clearly that the better performance of the original HL with respect to MAP$_{HL}$ is related to the GrIS sites being concentrated in a narrow window of annual mean temperature values.

Figure 7: legend is missing
The legend has been added, sorry about forgetting it in the previous version.

Supp Info section S2: the GrIS RMSE value for surface mass balance flux is quoted as 69 mm w.e. in Noël et al. (2019), not 69 m w.e. – check that the units have been correctly converted when applying random noise to the boundary conditions
This was a typo in the text. The value used in the random noise application was indeed 69 mm w.e. The text has been corrected (p.34 l.19 and p.35 l.5).

Supp Info section S2: equation S5 contains the term cn, should it be cp?
Yes, this has been corrected.

Supp Info section S2: '…must not be iteration specific…' – needs clarification
We changed the text to clarify our approach (p.35 l.20):
"In contrast to the climatic perturbation, the perturbation in $\rho_0$ can be specific to each single time step $t$, and the perturbation thus varies throughout the duration of a firn model simulation. Indeed, it is not unrealistic that a month with anomalously low fresh snow density is immediately followed by a month of anomalously high fresh snow density for example."

Supp Info section S2: please include a reference to justify the choice of 25 kg/m3 when defining the perturbation to the fresh snow density values
We have modified the text and included the reference to Reeh et al. (2005) (p.35 l.25):
"We determine surface density values at each site from the firn cores of our dataset, $\rho_0^{core}$, and we perturb these values based on a standard deviation of 25 kg m$^{-3}$. This value goes in line with a typical window of local variability of 50 kg m$^{-3}$ for $\rho_0$ (Reeh et al., 2005)."
The relevant sections in Reeh et al. (2005) are Section 4.2 and Figure 2. It should be noted that observations of fresh snow density variability in time but at a same location are scarce. However, it is this value that the perturbations in $\rho_0$ should represent. In contrast, observations of fresh snow density variability in space are more common. Therefore, we based our choice on the data of Reeh et al. (2005) that show the variability in $\rho_0$ at sites of similar average temperature. We believe that this variability better suits our interest than an ice sheet scale variability (e.g. Fausto et al., 2018).

Supp Info section S3: please clarify that 'original values' refers to parameter values from the original publications of the HL and Ar models
Changed (p.36 l.4):

"where $HL$ and $Ar$ subscripts denote the values from the original publications of the HL and Ar models, and the $mv$ subscript denotes a modified value of the parameter."

Supp Info section S6: start of second-to-last sentence – clarify what 'it' refers to

5   We clarified the subject of the sentence (p.37 l.35):
"As a consequence, the normal approximation results in a slight overestimation of uncertainty and thus conservative estimates of uncertainty."

Supp Info section S7: second sentence should refer to figure S5

10   Thank you, this has been modified (p.38 l.4).

Supp Info, Table S1: Please clarify whether accumulation and temperature values are taken from original publications or RACMO2
We added in the caption: "Values for both $\dot{b}$ and T are computed from the RACMO2 model."

We also let you know that a "References" section has been added at the end of the Supplementary Information, with the four references used. Three of these references are also cited in the main manuscript.

[revised manuscript text omitted]

In order to let uncertainty in RACMO2 output affect the calibration process, we perturb the temperature and accumulation time series that serve as climatic forcing for the firn models. At each iteration (a round of simulations with a given parameter set at all the calibration sites) and for each individual calibration site, we randomly draw an individual climatic perturbation value $c_p$ from a standard Normal distribution (Eq. (S1)). As such, every calibration site has a specific $c_p$ value, which changes at each iteration. We use observed statistics of RACMO2 errors in temperature and Surface Mass Balance to determine the perturbation. For GrIS, Noël et al. (2019) report RMSE values with respect to field observations for temperature and surface mass balance flux of 2.1 K and 69 mm w.e. yr$^{-1}$ respectively (in their Supplementary Material). Each monthly value of the RACMO2 time series is therefore perturbed by the corresponding RMSE value scaled by $c_p$ (Eq. (S2), (S3), (S4)).

We favour this approach rather than drawing a different random perturbation at each time step. The latter method would cause perturbations of opposite signs to occur on a very short timescale, which would result in unrealistic short term climatic variability (e.g. a very warm perturbation could be immediately followed by a very cold perturbation in the next month). Also, using the same $c_p$ value to quantify the magnitude of the perturbation for temperature and accumulation preserves the strong correlation between these variables. Warm (cold) temperature perturbations coincide with high (low) accumulation perturbations, which keeps our random perturbations physically plausible.

The part of the total accumulation perturbation attributed to each monthly time step is weighted by the actual accumulation at that time step. This attributes larger absolute noise in accumulation to high-accumulation months and lower absolute noise to low-accumulation months (Eq. (S3), (S4)).

Our approach is summarized in Eq. (S1), (S2), (S3) and (S4). These equations are applied at all iterations of the calibration process.

$$c_p \sim N(0,1) \quad \text{at all calibration sites} \quad \text{(S1)}$$
$$T_t^* = T_t + c_p \sigma_T \quad \text{at all } t \quad \text{(S2)}$$
$$\dot{b}_{tot}^* = n_{yr} c_p \sigma_{SMB} \quad \text{(S3)}$$
$$\dot{b}_t^* = \dot{b}_t + \dot{b}_{tot}^* \frac{\dot{b}_t}{\sum_t \dot{b}_t} \quad \text{(S4)}$$

where $T_t$ and $\dot{b}_t$ are temperature and accumulation rate as computed by RACMO2 at time step $t$ and the $*$ superscript denotes the perturbed quantity. $n_{yr}$ is the total number of years in a given simulation, $\dot{b}_{tot}^*$ is the total accumulation perturbation applied for that simulation and $\sigma_T$ and $\sigma_{SMB}$ are the temperature and surface mass balance flux RMSE values (as mentioned above, $\sigma_T = 2.1$ K and $\sigma_{SMB} = 69$ mm w.e. yr$^{-1}$ for GrIS). Note that by using a RMSE value on the surface mass balance flux, we overestimate uncertainty because the observed RMSE is mostly driven by errors in melt amounts which do not apply at the sites of our dataset, all from the dry snow zone area. For AIS, we apply the exact same process for perturbing the temperature variables. We use the RMSE value reported by van Wessem et al. (2018) and set $\sigma_T = 1.3$ K. The accumulation conditions of AIS forces the use of a slightly different method for perturbing the accumulation rate. In terms of magnitude, RACMO2 errors are much larger in coastal areas, where accumulation rates are high. In contrast, in the dry interior of the ice sheet where most of the cores of our dataset come from, the magnitude of RACMO2 errors is small due to low accumulation rates. As such, applying noise based on the ice sheet wide RMSE value would result in noise signals larger than actual accumulation values at most of our dry sites. We thus use the average RACMO surface mass balance bias of 5% (van Wessem et al., 2018) as a proxy for one standard deviation. For AIS, Eq. (S3) and (S4) are replaced by Eq. (S5).

$$\dot{b}_t^* = \dot{b}_t + 0.05\, c_{pn}\, \dot{b}_t \quad \text{(S5)}$$

As explained in Sect. 2.2, we also let uncertainty in fresh snow density, $\rho_0$, affect the calibration process by applying random perturbations to each $\rho_0^t$. In contrast to the climatic perturbation, the perturbation in $\rho_0$  can be specific to each single time step $t$, and the perturbation thus varies throughout the duration of a firn model simulation. Indeed, it is not unrealistic that a month with anomalously low fresh snow density is immediately followed by a month of anomalously high fresh snow density for example. We determine surface density values at each site from the firn cores of our dataset, $\rho_0^{core}$, and we perturb these values based on a standard deviation of 25 kg m$^{-3}$. This value goes in line with a typical window of local variability of 50 kg m$^{-3}$ for $\rho_0$ (Reeh et al., 2005). As such, adding noise to $\rho_0$ simplifies to Eq. (S6).

$$\rho_{0,t}^* \sim N(\rho_0^{core}, 25) \quad \text{(S6)}$$

We emphasize that the aim of this study is not to conduct a complete sensitivity analysis of firn densification to climatic forcing and to fresh snow density. The objective of the perturbations is to let reasonable estimates of errors in those fields to be accounted for in the calibration process.

**S3 Prior correlations in HL and Ar**

The Arrhenius form of HL and Ar (Eq. (4) and (5)) allows us to include some correlation in the prior distributions over the parameters of these models. The values of the Arrhenius pre-exponential factors ($k_0^*$, $k_1^*$, $k_0^{Ar}$ and $k_1^{Ar}$) are correlated with their corresponding activation energies ($E_0$, $E_1$ and $E_g$). For any given constant temperature, modelled densification rates, $\frac{d\rho}{dt}$, can be kept constant despite a change in the pre-exponential factor if the corresponding activation energy is changed accordingly and vice versa. As such, changes in these parameters can potentially compensate in the calculation of $DIP$ values and of the likelihood function (Eq. (8)). By enforcing constant $\frac{d\rho}{dt}$, exact compensation is ensured by the following equalities:

$$k_{0,mv}^* = k_{0,HL}^* \exp\left(\frac{E_{0,mv} - E_{0,HL}}{R\,T}\right) \quad \text{(S7)}$$

$$k_{1,mv}^* = k_{1,HL}^* \exp\left(\frac{E_{1,mv}-E_{1,HL}}{R\,T}\right) \quad \text{(S8)}$$

$$k_{0,mv}^{Ar} = k_{0,Ar}^{Ar} \exp\left(\frac{E_{g,Ar}-E_{g,mv}}{R\,T}\right) \quad \text{(S9)}$$

$$k_{1,mv}^{Ar} = k_{1,Ar}^{Ar} \exp\left(\frac{E_{g,Ar}-E_{g,mv}}{R\,T}\right) \quad \text{(S10)}$$

where $HL$ and $Ar$ subscripts denote the  values from the original publications of the HL and Ar models,

5    and the $mv$ subscript denotes a modified value of the parameter. Firstly, we generate 10000 random values of temperature $T$ in the range of annual mean temperatures covered by our dataset. Secondly, for each random temperature, we generate random values of $E_{0,mv}$, $E_{1,mv}$ and $E_{g,mv}$ in an interval of $\pm500$ J mol$^{-1}$ around the original values. Thirdly, we calculate the corresponding values in the pre-exponential factors from Eq. (S7), (S8), (S9) and (S10). This results in 10000 pairs of $(k_{0,mv}^*, E_{0,mv})$, $(k_{1,mv}^*, E_{1,mv})$, $(k_{0,mv}^{Ar}, E_{g,mv})$ and $(k_{1,mv}^{Ar}, E_{g,mv})$,

10   from which we calculate correlation coefficients. The absolute values of all four correlation coefficients lie in the interval [0.75; 0.78]. We decide to fix all prior correlation coefficients to -0.75 (HL parameters, negatively correlated) and 0.75 (Ar parameters, positively correlated). The process described necessarily results in perfectly correlated $k_{0,mv}^{Ar}$ and $k_{1,mv}^{Ar}$. We also set the prior correlation between these parameters to 0.75.
We emphasize here that any other pair of *a priori* uncorrelated parameters can certainly be correlated *a posteriori*

15   if the calibration process identifies such quantitative behaviour when the observed data is considered. This is highlighted and further discussed in Sect. S7.

**S4 The likelihood function, Eq. (8)**

[revised manuscript text omitted]

**S6 Normal approximation to the posterior**

The ensembles of parameter combinations obtained for each model provide large samples, representative of the

25   posterior probability distributions over their respective parameter space. The most efficient way to assess parameter-related uncertainty is to run a model with a high number of random parameter combinations from these ensembles, which is demonstrated in Sect. 3. However, this means that for any firn modelling study, access must be easy to such posterior ensembles or an MCMC algorithm must be re-executed. To circumvent these practical difficulties, it is approximately correct to sample random parameter combinations from a multivariate normal

30   distribution centred about the mean of the posterior ensemble and with covariance matrix set to the posterior ensemble covariance matrix. This is commonly referred to as a normal approximation to the posterior (Gelman et al., 2013). Table S2 provides both the posterior mean and posterior covariance for the HL, Ar and LZ models. We assess how random samples from the normal approximations compare to samples from the posterior ensembles in Fig. S4. Posterior samples and the normal approximations are very similar, with correlations only

35   slightly less well captured in the tails of the distributions.  As a consequence, the normal approximation results in a slight overestimation of uncertainty and thus conservative estimates of uncertainty. This has been confirmed by additional model simulations with values sampled from the normal approximations (not shown).

**S7 Posterior correlation between parameters**

[revised manuscript text omitted]

**Table S1.** The 91 firn core dataset used in this study. * symbols indicate the core is part of the evaluation data. Lat and Lon designate latitude and longitude respectively. Year indicates the year of drilling of the core. $\dot{b}$ is the accumulation rate. T is the temperature. Values for both $\dot{b}$ and T are computed from the RACMO2 model. $\rho_0$ is the surface density boundary condition that was derived individually for each core by extrapolating density measurements until the surface (random noise is added to $\rho_0$ as discussed in Sect. S2). Var designates the site-specific variance used for the terms of $\Sigma_{15}$ and $\Sigma_{pc}$ (see Text S4 for their calculation). The core spencer90 has only a single density measurement above 15 m depth and its $DIP15$ is discarded.

| | Parameters | Posterior mean | Posterior covariance matrix |
|---|---|---|---|
| HL | $k_0^*, k_1^*, E_0,$ $E_1, a, b$ | $\begin{bmatrix} 16.7, 649, 10760, \\ 21000, 0.88, 0.66 \end{bmatrix}$ | $\begin{bmatrix} 34.4 & 40.2 & 4500 & 324 & -0.0685 & -0.0195 \\ 40.2 & 44000 & 618 & 161000 & 1.087 & -3.670 \\ 4502 & 618 & 710000 & 7080 & -29.95 & 1.94 \\ 324 & 1610000 & 7080 & 694000 & 7.86 & -27.51 \\ -0.0685 & 1.087 & -29.95 & 7.86 & 0.0051 & -0.0012 \\ -0.0195 & -3.670 & 1.94 & -27.51 & -0.0012 & 0.0036 \end{bmatrix}$ |
| Ar | $k_0^{Ar}, k_1^{Ar}, E_g,$ $\alpha, \beta$ | $\begin{bmatrix} 0.080, 0.028, 40900, \\ 0.78, 0.69 \end{bmatrix}$ | $\begin{bmatrix} 5.62 \times 10^{-4} & 1.55 \times 10^{-4} & -12.66 & 9.65 \times 10^{-5} & -3.23 \times 10^{-4} \\ 1.55 \times 10^{-4} & 7.41 \times 10^{-5} & -4.64 & -2.04 \times 10^{-4} & 1.05 \times 10^{-4} \\ -12.66 & -4.64 & 360000 & 11.0 & 4.67 \\ 9.65 \times 10^{-5} & -2.04 \times 10^{-4} & 11.0 & 0.00330 & -0.00101 \\ -3.23 \times 10^{-4} & 1.05 \times 10^{-4} & 4.67 & -0.00101 & 0.00312 \end{bmatrix}$ |
| LZ | $lz_a, lz_b, lz_{11},$ $lz_{12}, lz_{13}, lz_{21},$ $lz_{22}, lz_{23}$ | $\begin{bmatrix} 7.56, -2.091, -14.71, \\ 7.269, -1.019, -1.513, \\ 6.0203, -0.09127 \end{bmatrix}$ | $\begin{bmatrix} 5.27 & -0.198 & -1.20 & -1.68 & -0.0239 & 0.00553 & -0.0606 & 0.00413 \\ -0.198 & 0.0116 & 0.218 & -0.0612 & 0.0134 & -0.0158 & -0.00229 & -7.37 \times 10^{-4} \\ -1.20 & 0.218 & 14.6 & -3.96 & 0.801 & 0.368 & 0.354 & 0.0129 \\ -1.68 & -0.0612 & -3.96 & 13.3 & -0.309 & -0.0850 & 5.40 & 0.0166 \\ -0.0239 & 0.0134 & 0.801 & -0.309 & -0.0502 & -0.0173 & 0.0252 & -4.42 \times 10^{-4} \\ 0.00553 & -0.0158 & 0.368 & -0.0850 & -0.0173 & 0.446 & -0.429 & 0.0131 \\ -0.0606 & -0.00229 & 0.354 & 5.40 & 0.0252 & -0.429 & 3.94 & -2.59 \times 10^{-4} \\ 0.00413 & -7.37 \times 10^{-4} & 0.0129 & 0.0166 & -4.42 \times 10^{-4} & 0.0131 & -2.59 \times 10^{-4} & 4.80 \times 10^{-4} \end{bmatrix}$ |

**Table S2.** The posterior means and covariance matrices for the free parameters of HL, Ar and LZ. These statistics can be used to generate random parameter combinations following a normal approximation.

**Figures**

[Figure]

**Figure S1.** Climatic conditions at the 91 sites of the dataset

[Figure]

**Figure S2.** Quantiles-Quantiles plots for the errors of the three original models (HL, Ar, LZ) computed on the entire dataset. The alignment of the points along the red line informs about the fit to a normal distribution.

[Figure]

**Figure S3.** Sampling chains of each parameter for (a) HL, (b) Ar, (c) LZ. The x-axis displays the iteration number, the y-axis displays the parameter value. The dashed pink line shows the value of the original model, which is also the starting point of each chain.

[Figure]

**Figure S4.** Evaluation of the normal approximations to the posterior distributions for (a) HL, (b) Ar, (c) LZ. Where possible, correlated parameters share a same graph.

[Figure]

**Figure S5.** Posterior correlation matrices.

[Figure]

**Figure S6.** Comparison of evaluation data *DIP* with model results for the LZ dual and IMAU models.

---

## Editor Decision (ED2)

**Editor comments on "Bayesian calibration of firn densification models" by Verjans et al.**

I would like to thank the authors for providing a thorough, detailed response to the first round of reviewer comments and for the steps they have taken to demonstrate how their results are of practical use to the wider community of firn model users. In their rebuttal document the authors provide detailed insight into the decisions made when carrying out this study and responses to all reviewers' comments are clearly justified.

Following recommendations from the reviewers, the authors have undertaken additional simulations that include a more robust treatment of uncertainty on input parameters, an appreciation of correlation in the prior distribution of model parameters, and a clearer analysis of the spread in the results. I support the decision to remove the section that seeks to apply the results to the whole of the Greenland Ice Sheet - the authors provide very clear justification that a robust analysis would require consideration of factors that are outside the scope of this study.

Reviewer 1 is happy with the edits that have been made to the manuscript. I have carried out a review of the current version of the manuscript with the aim of (i) checking that issues raised by reviewer 2 have been addressed and (ii) assessing whether the article is now ready for publication.

All technical aspects of the article appear to be robust and there is no need for any additional analysis to be carried out. However, I list below a number of minor points that should be addressed before the article can be accepted for publication. These either seek clarification on specific issues or are suggestions to improve the clarity of the text. I am not an expert in firn modelling, and some of the points may reflect gaps in my knowledge, but they are made with the aim of ensuring the article is accessible to the non-specialist. All points relate to minor text/presentation issues that should be relatively straightforward to implement. Once they are addressed, I would be happy to accept this article for publication in The Cryosphere.

Kind regards,

Pippa Whitehouse
Associate Editor, The Cryosphere

Minor points – page/line numbers relate to version2 of the manuscript (no track changes)

p.1 l.31: most points in this paragraph document why each application is important rather than recommending specific steps for improvement – the text on line 31 is a little misleading since it could be taken to suggest that you will investigate sensitivity to climate conditions in this study

p.2 l.16: check throughout whether it is appropriate use to use 'AIS/GrIS' or 'the AIS/GrIS'

p.4 l.5-7: text seems a little out of place here, information would fit better at the end of the final paragraph of p.3

p.4 l.25-26: the statement that you use a constant site-specific value is slightly at odds with the statement on lines 29-30 that you add random noise at every model time step. It would be useful to mention the approach used to account for uncertainty earlier in this paragraph

p.6 l.9-11: do Li and Zwally (2015) use a different formulation of the equation, or do they just determine different parameter values compared with Li and Zwally (2011)?

p.6 l.25: to data -> with data

p.7 l.23: several steps are described prior to the mention of figure 2a; please indicate how the text on lines 20-23 relates to steps shown in figure 2. Also, please clarify whether calculations are carried out for both $\theta_i$ and $\theta_i^*$ on each iteration

p.8 l.1: variance -> covariance (as defined on l.25 of the previous page)

p.8 l.16: 'a 500 random sample' – rephrase

p.9 l.22: the DML plots are figs. 5g-i

p.9 l.28: 'the better performance at the GrIS evaluation sites…' – make it clear that this text relates to the performance of the original model

p.10 l.7: it is not clear to me how the LZ dual model was constructed; do you determine different parameters for each ice sheet by dividing the calibration data set, or is the whole formulation of the model different? Refer to Table 2 when quoting results for the LZ dual model

p.10: if feasible, it would be useful to include a figure showing the results for LZ dual and IMAU-FDM (e.g. similar to figure 4) in the supplementary information

p.10 l.23: please include information on how uncertainty intervals were constructed in the captions to figures 4 and 5

p.11 l.5: 'indicate a weaker increase…' – weaker than what?

p.11 l.12: 'The same can be applied…' – not clear what 'The same' refers to

p.11, l.13: please refer to a figure or table when quoting correlation coefficient values

p.11 l.22: over-sensitivity -> over-sensitivity in Ar

p.12 l.25: what is the reference period? If it is 2000-2017 this should be explicitly stated

p.12 l.33: make it clearer that uncertainties in the following sentences are calculated using the CV values quote above/ in table 3

p.13 l.5: a couple of clarifications needed: (i) what does 'it' refer to, and (ii) what does 'Such numbers' refer to?

p.13 l.7: the purpose of the text (paragraph?) starting on this line is initially unclear. For example, it is not clear what you mean by 'the different sensitivities…'. You mention that compaction is sensitive to variability and 'general increases' in temperature and accumulation – can you be more explicit about the climate at the two sites, perhaps by including site-specific RACMO2 output in figure 7?

p.13 l.14: short-scale -> short-timescale

p.13 l.26: 'at most sites…uncertainty intervals do not cover observed DIP values' – this is an important result but I did not see it stated/quantified anywhere in the main text

p.14 l.12-13: this link takes you to a folder which contains several files that are unrelated to this study, are you able to list a source that just links to the firn data?

Tables 1 and S2: some terminology issues for large/small values, e.g. $9\ 10^4$ should be $9 \times 10^4$

Table 2: explicitly mention RMSE in the caption rather than just 'The errors'

Figure 1: what is the difference between a circle and a cross?

Figure 2: in the box titled 'If i is multiple of 100' the second $\Sigma$ should be $\Sigma_{cov}$

Figure 3: please document what the 'posterior samples' are. Do they represent parameters associated with the 500 parameter sets randomly selected from the ensemble of accepted θ?

Figure 6: mention the difference between the left and right columns in the caption. Also, is it possible to represent AIS and GrIS data points differently, to support statements in main text?

Figure 7: legend is missing

Supp Info section S2: the GrIS RMSE value for surface mass balance flux is quoted as 69 mm w.e. in Noël et al. (2019), not 69 m w.e. – check that the units have been correctly converted when applying random noise to the boundary conditions

Supp Info section S2: equation S5 contains the term $c_n$, should it be $c_p$?

Supp Info section S2: '…must not be iteration specific…' – needs clarification

Supp Info section S2: please include a reference to justify the choice of 25 kg/m$^3$ when defining the perturbation to the fresh snow density values

Supp Info section S3: please clarify that 'original values' refers to parameter values from the original publications of the HL and Ar models

Supp Info section S6: start of second-to-last sentence – clarify what 'it' refers to

Supp Info section S7: second sentence should refer to figure S5

Supp Info, Table S1: Please clarify whether accumulation and temperature values are taken from original publications or RACMO2

---

## Editor Decision (ED3)

Dear authors,

Thank you for your thorough and efficient responses to all points raised during the review process. This is a novel study into firn densification modelling and one which provides important quantification of uncertainties associated with model choice and model parameters via a Bayesian approach. I am happy to approve this article for publication in The Cryosphere.

Thank you for choosing to publish your research in The Cryosphere.

Pippa Whitehouse

Associate Editor, The Cryosphere